# Understand and Accelerate Memory Processing Pipeline for Large Language Model Inference

**Zifan He** [1]  **Rui Ma** [2]  **Yizhou Sun** [1]  **Jason Cong** [1]

## Abstract

Modern large language models (LLMs) increasingly depend on efficient long-context processing and generation mechanisms, including sparse attention, retrieval-augmented generation (RAG), and compressed contextual memory, to solve complex tasks. We show that these optimizations can be unified into a four-stage memory processing pipeline: *Prepare Memory*, *Compute Relevancy*, *Retrieval*, and *Apply to Inference*. Through systematic profiling, we identify a 22%-97% memory processing overhead in LLM inference and strong computational heterogeneity across stages in memory processing. Motivated by this insight, we argue that **heterogeneous systems** are well-suited to accelerate memory processing and thus end-to-end inference. We demonstrate this approach on a GPU-FPGA system by offloading sparse, irregular, and memory-bounded operations to FPGAs while retaining compute-intensive operations on GPUs. Evaluated on an AMD MI210 GPU and an Alveo U55C FPGA, our system is up to $2.2\times$ faster and $4.7\times$ energy reduction across multiple LLM optimizations than the GPU baseline (with similar results on NVIDIA A100), establishing heterogeneous systems as a practical direction for efficient LLM inference and informing future heterogeneous hardware design.

## 1. Introduction

In recent years, large language models (LLMs) have demonstrated capabilities beyond the simple question-answering tasks. LLM-based agents can now solve complex tasks through multi-step reasoning (Yao et al., 2023a; Wang &

[1]Department of Computer Science, University of California, Los Angeles [2]Microsoft Research Asia. Correspondence to: Zifan He <zifanhe1202@g.ucla.edu>, Rui Ma <mrui@microsoft.com>.

*Proceedings of the 43rd International Conference on Machine Learning*, Seoul, South Korea. PMLR 306, 2026. Copyright 2026 by the author(s).

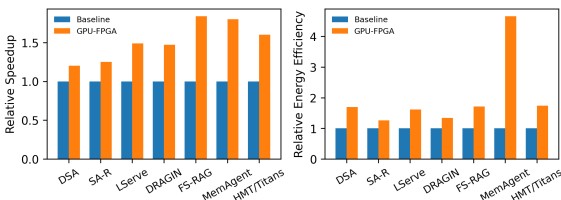

*Figure 1.* The GPU-FPGA heterogeneous system (1 MI210 + 1 Alveo U55C) can provide $1.2-1.8\times$ speedup and $1.3-4.7\times$ energy cost reduction consistently over a wide range of long-context LLM inference optimizations. "SA-R" stands for SeerAttention-R and "DSA" stands for DeepSeek Attention.

Zhou, 2024), tool invocation (Yao et al., 2023b), and long-horizon planning (Wang et al., 2023), which increasingly demands the ability to memorize and process long inputs. State-of-the-art models (Comanici et al., 2025; Shen et al., 2025; Grattafiori et al., 2024) can process and generate 128k to 1 million tokens per request when prompted for paper reading, deep reasoning, and creative writing. Standard LLMs maintain all contexts as key-value (KV) caches, incurring substantial hardware costs and runtime overhead. For example, storing KV cache for 1M tokens requires up to 69 GB of GPU memory for the GPT-OSS-120B model (Agarwal et al., 2025), and repeatedly accessing the cache further amplifies the pressure on memory access during auto-regressive decoding. To mitigate this issue, the latest LLMs have developed several algorithmic optimizations to improve LLM memory efficiency. Representative approaches include sparse attention (Beltagy et al., 2020; Liu et al., 2025; Yang et al., 2025b) that selectively attends to a subset of tokens, and contextual memory (Behrouz et al., 2026; Sun et al., 2025) that compresses past tokens into embeddings. Additionally, retrieval augmented generation (RAG) (Lewis et al., 2020) can be used to offload static knowledge to an external database and, thus, considered as an approach to expand the context.

While these LLM inference optimizations effectively reduce the end-to-end latency of long document processing, prior work largely treats them as isolated techniques and lacks a systematic understanding of their computational characteristics and hardware efficiency implications. This deficiency hinders further acceleration of LLM inference for both existing and emerging methods. In this work, we make three

insightful claims that will open a new way to understand and further accelerate the long document LLM inference.

**Claim 1: Modern LLM inference involves a memory processing pipeline. (Section 3)** Existing LLM inference optimizations mostly address the challenge of efficient memory processing. We formally define *memory*[1] in LLMs and unify diverse methods under a common four-step pipeline: (1) *Prepare Memory*, which preprocesses raw memory into a compact or structured format; (2) *Compute Relevancy*, which assigns importance scores to each memory entry; (3) *Retrieval*, which selects and extracts information based on these scores using specific heuristics; and (4) *Apply to Inference*, which integrates the retrieved content into the decoding process. Through profiling, we observe that memory processing accounts for $22\% - 97\%$ of the total latency. These findings not only reveal substantial opportunities for accelerating LLM, but also highlight that *a single system solution targeting memory processing can provide benefits across existing and future LLM inference paradigms.*

**Claim 2: Computations in memory processing are heterogeneous. (Section 4)** Quantitatively, each step exhibits distinct arithmetic intensity, leading to different utilization of compute and memory resources. Qualitatively, their memory access patterns and data dependencies also differ substantially. For example, generating compressed key embeddings in sparse attention involves regular, consecutive accesses and is compute intensive when processing multiple attention heads, whereas score computation and retrieval rely on skinny matrix-vector multiply and top-$k$ search that are memory-bound, irregular in access pattern, and data-dependent across tokens. We observe that such computational heterogeneity is pervasive in the memory processing pipeline of LLM inference.

**Claim 3: Heterogeneous systems can accelerate memory processing. (Section 5)** Motivated by the heterogeneity in computational characteristics, we argue that mapping LLM memory processing onto a heterogeneous system is preferred to achieve optimal acceleration. In this work, we present a solution based on off-the-shelf devices for demonstration. The paradigm can be used to inspire future heterogeneous hardware designs. In particular, given the speed and energy advantages of Field Programmable Gate Array (FPGA) devices for sparse, irregular, and memory-bound workloads over GPUs and CPUs (Song et al., 2022; He et al., 2024), we propose executing the memory processing pipeline of LLM inference on a GPU-FPGA heterogeneous system, with consideration of both computational heterogeneity and data locality. We evaluated a system with an AMD MI210 GPU and an Alveo U55C FPGA connected via PCIe, and it achieves:

---

[1]In this work, "memory" refers to the processed data the LLMs used for generation in Definition 3.1 (not the device memory).

- $1.5 \sim 5.7\times$ speedup for the sparse attention (e.g., DeepSeek Sparse Attention (Liu et al., 2025)), resulting in up to $1.49\times$ faster end-to-end inference.
- $5.2 \sim 7.7\times$ speedup for RAG (e.g., DRAGIN (Su et al., 2024)), resulting in up to $2.2\times$ end-to-end speedup.
- $1.3 \sim 1.6\times$ end-to-end speedup for Memory as Context, and $1.8\times$ for synthesized memory (MemAgent).
- $1.1 \sim 4.7\times$ lower geomean energy cost per request, which can significantly reduce the serving cost of LLMs equipped with these methods. [2]

**Conflict of Interest Disclosure.** The author Jason Cong has a financial interest in AMD, which develops the AMD MI210 GPU and U55C FPGA evaluated in this paper.

## 2. Background

We review representative long context/document LLM inference optimizations, focusing on methods that fundamentally change how memory is accessed and managed.

**Sparse Attention.** Standard transformers incur quadratic complexity during prefill (input processing) and linear complexity during decoding (token generation) (Sheng et al., 2023), making attention increasingly expensive for long contexts. Sparse attention mitigates this by selectively attending to a subset of past tokens. Recent decode-stage methods include DeepSeek Attention (Liu et al., 2025), which retrieves top-$k$ important tokens with a lightweight indexer, LServe (Yang et al., 2025b), introducing a hierarchical paged KV cache for fast retrieval, and SeerAttention-R (Gao et al., 2026a), which extends an auxiliary attention predictor to the decoding phase. These approaches significantly reduce memory access overhead while preserving model quality.

**Retrieval-Augmented Generation (RAG).** For static knowledge sources, storing information as KV cache is inefficient due to per-layer vector storage. RAG (Lewis et al., 2020) instead gets relevant documents from an external corpus and concatenates them with the query. Representative dynamic RAG systems include FLARE (Jiang et al., 2023), which triggers retrieval when model confidence drops, and DRAGIN (Su et al., 2024), which leverages attention statistics to detect uncertainty. Fixed-sentence RAG (Trivedi et al., 2023) performs retrieval at every sentence boundary, enabling fine-grained context updates. Recently, more advanced two-stage RAG with reranker (Moreira et al., 2024) improve the accuracy by sacrificing the retrieval overhead. These methods enhance the precision of retrieving factual knowledge while reducing long-context storage overhead.

**Compressed Contextual Memory.** Compressed memory stores information in a compact form (embeddings or summarized texts), enabling LLMs to handle extremely long

---

[2]https://github.com/OswaldHe/HeteroLLM.

*Table 1.* Summary of LLM inference optimizations and the computations in their memory processing pipeline.

| LLM Opt. | Related Works | Prepare Memory | Compute Relevancy | Retrieval | Apply to Inference |
|---|---|---|---|---|---|
| Sparse Attention | DeepSeek Attention (Liu et al., 2025) | Linear Projections + RoPE | Multi-headed Inner Product | Top-$k$ | Fine-grain Sparse Attention |
| | SeerAttention-R (Gao et al., 2026a) | Linear Projections + Pooling | Inner Product | Top-$k$ / Threshold | Block Sparse Attention |
| | LServe (Yang et al., 2025b) | Page-wise Min/Max Pooling | Inner Product + Max Reduction | Top-$k$ | Block Sparse Attention |
| Retrieval Augmented Generation (RAG) | Two-stage RAG (Moreira et al., 2024) | Embedding Model / Tokenization | Inner Product / BM25 + Reranker | Top-$k$ | Append to query |
| | Fixed-Sentence RAG (Trivedi et al., 2023) | Tokenization | BM25 | Top-$k$ | Append to query |
| | Dynamic RAG (Su et al., 2024; Jiang et al., 2023) | Tokenization | BM25 | Top-$k$ | Append to query |
| Synthesized Memory | MemAgent (Yu et al., 2026; Shen et al., 2025) | Model Decoding | N/A | Nearest Retrieval | Model Prefilling |
| Memory as Context | Titans (Behrouz et al., 2026), HMT (He et al., 2025a) | Forward pass | Linear Projection + Inner Product | Top-$k$ / Weighted Sum | Append to segment |
| Test-time Training | TTT, LaCT (Sun et al., 2025; Zhang et al., 2026b) | Backward pass | Compute loss | N/A | Forward pass |

inputs. Synthesized memory such as MemAgent (Yu et al., 2026) summarizes long inputs into textual memories conditioned on the query. Recurrent models, including HMT (He et al., 2025a) and Memory as Context in Titans (Behrouz et al., 2026), compress input segments into latent embeddings and retrieve them based on relevancy. These methods effectively trade computation for runtime data efficiency and assist far-distance information retrieval in the latent space.

**Test-time Training (TTT).** TTT treats model parameters as internal memory and adapts them during inference. The seminal work (Sun et al., 2025) formulates TTT as a recurrent update rule, alternating between backpropagation and generation. LaCT (Zhang et al., 2026b) extends this idea with batched updates to improve GPU utilization.

Although these methods differ in their computations, they all transform a processed input into an output through a common sequence of steps referred to as **memory processing**.

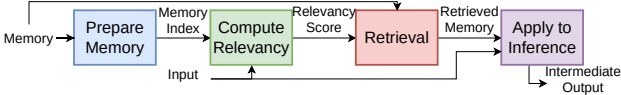

*Figure 2.* **Four-Step Memory Processing Pipeline in LLMs:** *Prepare Memory* preprocesses and structures raw memory for efficient access; *Compute Relevancy* assigns relevance scores to memory entries with respect to the input query; *Retrieval* extracts the most relevant memory based on these scores; and *Apply to Inference* integrates retrieved content and input into intermediate outputs, used in the rest operations in LLMs to produce tokens.

## 3. Memory Processing in LLM Inference

**Definition 3.1.** Define the generative language model as $L(g(\cdot), f(\cdot, \cdot), \{x_i\}_{i<t}, x_t) = y_t$, where $\{x_i\}_{i<t}$ is the past input sequence, $x_t$ and $y_t$ are current input and output, $g$ is the memory generator, and $f$ is the memory processor. $L$ starts with generated memory $M_{<t} = g(\{x_i\}_{i<t})$. Then during inference, it initiates $f$ (one-time or repeatedly) to get intermediate output $O_{<t} = f(M_{<t}, x_t)$ and utilize $O_{<t}$ to generate final output $y_t$.

For example, $g$ can be the projections to produce KV cache $M_{<t}$, and $f$ is the sparse attention mechanism to create attention score $O_{<t}$. Under this definition, the methods mentioned in Section 2 are all improving the memory processing ($f$) efficiency. This section describes the common properties of the memory processing.

### 3.1. Memory Processing Pipeline

To utilize models' memory, LLM inference employs a four-stage memory processing pipeline (Figure 2):

- *Prepare Memory* ($\text{prep}(M_{<t}) = I_{<t}$): LLM first converts the memory into a memory index that facilitates memory retrieval and usage. For instance, DeepSeek attention projects latent KV vectors in MLA (multi-headed latent attention) (Liu et al., 2024) into lightweight indexing vectors.

- *Compute Relevancy* ($\text{comp}(I_{<t}, x_t) = S$): In this step, the model utilizes the processed memory and current input, or query, to identify which part of the memory is relevant and should be extracted. The output is the relevancy scores where higher score indicates higher relevancy. For example, DeepSeek attention computes the multi-head dot product scores between the indexing vectors and the query vector.

- *Retrieval* ($\text{ret}(M_{<t}, S) = M'_{<t}$): Given the relevancy score and the original memory, the model selects a subset of memory entries or constructs refined memory based on

the score and certain heuristics. For example, DeepSeek attention applies top-$k$ selection on each token.

• *Apply to Inference* (apply$(M'_{<t}, x_t) = O_{<t}$): Finally, the retrieved memory is incorporated into subsequent computations alongside the target inputs that the model transforms to generate the output. For instance, the KV latent embeddings corresponding to the top-$k$ index scores are used in the MLA computations. For RAG, the selected documents are concatenated with the query to augment the inference.

Table 1 outlines the LLM inference method types with representative works and the detailed computations involved in each step of the memory processing pipeline. Some methods skip a few steps: 1) MemAgent (Yu et al., 2026) skips relevancy computation since it always retrieves memory from the preceding segment. 2) TTT does not perform retrieval and incorporates parameterized memory through a direct forward pass. Moreover, the timing and execution frequency of each step in the end-to-end inference vary across different memory types. RAG typically prepares memory once and repeatedly executes retrieval, while sparse attention processes memory for every token.

This unified abstraction for memory processing in LLMs serves as an analytical framework to characterize computational properties and guide systematic accelerations. When a stage is not required, it introduces no overhead, as data can bypass the stage without additional computation or control complexity. Furthermore, it supports reuse of kernels employed for each stage and scheduling in the pipeline, facilitating efficient deployments.

### 3.2. Memory Processing is Time Critical

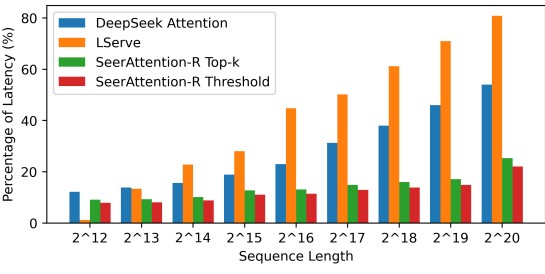

*Figure 3.* Percentage of latency spent on memory processing for sparse attention methods. With 1M tokens, memory processing can take 22%–81% of the decoding time.

By profiling the latency breakdowns of LLM inference optimizations based on the experimental settings illustrated in Section 6.1, we observe a large proportion of inference latency is spent on memory processing as the memory size increases. For sparse attention (Figure 3), the percentage of decoding latency for memory processing grows from $1 - 11\%$ for 4K-token sequences to $22 - 81\%$ for 1M-token sequences. Given that modern LLMs (Comanici et al., 2025)

can process more than 1M tokens per sample, memory processing can become the critical path when sparse attention is applied. For RAG (Figure 4), our profiling reveals a similar phenomenon when processing 20M documents ($40 - 61\%$). In two-stage RAG, the reranker dominates memory processing latency, leading to a high latency percentage with a slower increase as the document count grows. As depicted in Figure 5, for parameterized memory and synthesized memory, memory processing is time-consuming even with short contexts. The reasons are diverse: MemAgent employs the model to generate textual memory, and Titans and HMT involve multiple linear layers to project segment and memory embeddings into the same latent space.

Furthermore, we observe that the latency breakdown in the memory processing pipeline varies among methods. For example, sparse attention and RAG are dominated by the Compute Relevancy and Retrieval, whereas MemAgent incurs up to 97% of latency in Prepare Memory (details in Appendix B). Since memory processing constitutes a primary bottleneck in LLM, accelerating it delivers substantial gains in both end-to-end latency and energy efficiency.

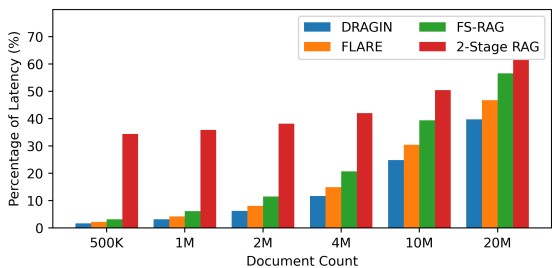

*Figure 4.* Percentage of latency on memory processing for RAG using the Wikipedia dump (Su et al., 2024). For two-stage RAG, reranking is time consuming, leading to a high percentage at 500K with a slow increment as the document count grows.

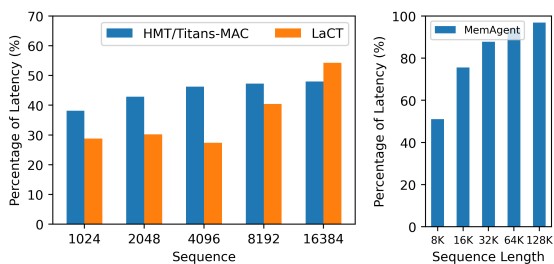

*Figure 5.* Left: Percentage of latency on memory processing for parameterized memory (Titans/HMT, LaCT). Right: Percentage of latency on memory processing for MemAgent.

## 4. Computational Heterogeneity

We further analyze the computational properties of the memory processing pipeline and reveal its heterogeneity. Quantitatively, arithmetic intensity (FLOPs/byte) (Williams et al.,

*Table 2.* Summary of LLM inference optimizations and computational properties of their memory processing pipelines for single batch. We show arithmetic intensity (FLOPs/byte, higher means more compute-bound) with only *orders of magnitude* (details in Appendix B). "Local Memory" denotes independent computation across memory entries, and "Regular" indicates consecutive data access.

| LLM Opt. | | Prepare Memory | Compute Relevancy | Retrieval | Apply to Inference | Rest Ops in LLM |
|---|---|---|---|---|---|---|
| | *Arith. Intensity* | 10–100 | 1–10 | 1 | 10–100 | 1–10 |
| Sparse Attention | *Access Pattern* | Regular | Regular | Irregular | Regular | Regular |
| | *Data Requirement* | Local Memory | Local Memory | Across Memories | Local Memory | Local Memory |
| | *Arith. Intensity* | 1–100 | 1–10 | 1 | 0 | > 100 |
| RAG | *Access Pattern* | Irregular (tokenizer) Regular (embeddings) | Irregular | Irregular | Regular | Regular |
| | *Data Requirement* | Local Memory | Across Memories | Across Memories | no calculations | Local Memory |
| | *Arith. Intensity* | 1–10 | - | 0 | > 100 | > 100 |
| Synthesized Memory | *Access Pattern* | Regular | - | Regular | Regular | Regular |
| | *Data Requirement* | Across Memories | - | No Calculation | Local Memory | Local Memory |
| | *Arith. Intensity* | > 100 | 1–10 | 1 | 0 | > 100 |
| Memory as Context | *Access Pattern* | Regular | Regular | Regular | Regular | Regular |
| | *Data Requirement* | Local Memory | Local Memory | Across Memories | Local Memory | Local Memory |
| | *Arith. Intensity* | > 100 | 1–10 | - | > 100 | > 100 |
| Test-time Training | *Access Pattern* | Regular | Regular | - | Regular | Regular |
| | *Data Requirement* | Local Memory | Local Memory | - | Local Memory | Local Memory |

2009) characterizes whether a step is memory-bound (frequent data access) or compute-bound (more arithmetics), while qualitatively, we examine data access patterns and dependencies. Table 2 summarizes these properties for each LLM inference optimization, with a detailed arithmetic intensity and latency breakdown provided in Appendix B.

*Sparse Attention & RAG*: Sparse attention and RAG exhibit similar heterogeneity across memory processing steps. Compute Relevancy and Retrieval are memory-bound (skinny matrix-matrix and matrix-vector multiplication) and involve irregular accesses and data dependencies, such as BM25 scoring, top-$k$ selection, and max reduction. For instance, BM25 lookups token frequency histograms in a non-deterministic order across documents, while top-$k$ maintains running maximum scores with data dependencies and irregular data eviction. In contrast, Prepare Memory and Apply to Inference are primarily dense linear algebra with consecutive and independent accesses, including linear projections and attention operations.

*Synthesized Memory*: The memory processing of MemAgent is essentially a sequence of LLM inferences. At the Prepare Memory step, the LLM decodes to generated textual memory, which is a memory-bound operation. At the Apply to Inference step, the LLM performs prefilling to consume the memory and the text from the current segment. This operation is compute-bound.

*Memory as Context*: Computations in Memory as Context are similar to sparse attention and RAG, except that there are extra calculations to generate the query from sequences in Compute Relevancy. These calculations are independent of the model forward pass and can be parallelized and fused with other kernels.

*TTT*: The heterogeneity is insufficient. Although computing the loss function (Compute Relevancy) is more memory-intensive than the other steps, the latency bottleneck of memory processing in LaCT is dominated by compute-bound operations (forward and backward pass). Thus, we do not deploy it on the heterogeneous system.

This heterogeneity of memory processing pipeline in LLM inference motivates the mapping of LLM onto a heterogeneous system as discussed in the next section.

# 5. GPU-FPGA Heterogeneous System

## 5.1. GPU vs. FPGA

GPUs are well-known for their efficiency in accelerating LLMs with abundant highly parallel compute cores and large HBM bandwidth. However, GPUs suffer from under-utilization of computational resources and off-chip memory bandwidth when processing irregular data accesses and memory-bound operations (Boutros et al., 2020; Song et al., 2022; Rajashekar et al., 2024; He et al., 2024). In contrast, FPGAs allow users to customize the microarchitecture for data control. The advantages of FPGAs over GPUs include: 1) larger SRAM capacity with higher bandwidth, 2) flexible data control with minimized scheduling efforts, and 3) low power consumption with competitive performance. These features highlight the potential of FPGAs to accelerate the memory processing pipeline of LLM inference, especially operations that are not hardware-friendly to GPUs.

## 5.2. Heterogeneous System Overview

As a demonstration, we consider a system with an FPGA and a GPU, both equipped with HBM and connected via PCIe. Our general mapping criteria for memory processing steps and data are: 1) deploying steps based on the strengths

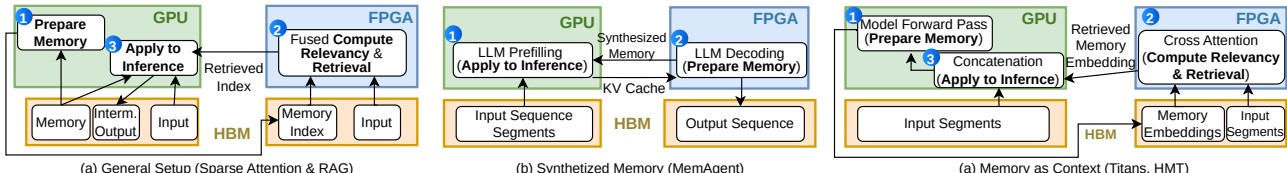

*Figure 6.* Kernel mapping and data communication on the GPU-FPGA system. (a) Sparse attention and RAG employ general setup, where the GPU prepares the memory and apply to the inference and the FPGA executes an efficient fused kernel for compute relevancy and retrieval. (b) For MemAgent, we utilize prefill-decode disaggregation where the FPGA operates LLM decoding and the GPU handles prefilling. (c) For Memory as Context, we update the data mapping for locality: memory on the FPGA, retrieved memory delivered from the FPGA to the GPU, and output is directly streamed to Prepare Memory.

of the FPGA and GPU (i.e., compute-bounded and regular data access on the GPU; irregular, data dependent, and memory-bound operations on the FPGA), and 2) balancing the trade-off between cross-device communication overhead and kernel-level speedup. We prioritize criterion 2 over 1 for minimal end-to-end latency. For example, although extracting the KV cache for top-$k$ tokens is memory bound, we do not fuse it with top-$k$ selection on the FPGA because the PCIe overhead outweighs the fusion benefit. Instead, we transfer only the top-$k$ indices to minimize PCIe latency and run KV cache extraction on the GPU. Figure 6 illustrates an overview of the mapping for different cases.

*General Setup*: This design applies for both sparse attention and RAG. We execute the Prepare Memory and Apply Memory steps on the GPU and deploy a fused Compute Relevancy and Retrieval kernel on the FPGA (Figure 6(a)). The GPU HBM stores the memory, target data, and output data, while the FPGA saves the query and the processed memory generated by the GPU. In sparse attention, the processed memory is a compressed KV cache. The GPU transfers the compressed KV cache for the entire input sequence during prefilling and only the next token during decoding. In RAG, Prepare Memory is a one-time process and amortized across subsequent steps. After retrieval, the FPGA returns the retrieved indices to the GPU for memory access.

*Synthesized Memory*: Similar to other works on prefill-decode disaggregation between the GPU and the FPGA (Yang et al., 2024b), we deploy Prepare Memory (LLM decoding) to FPGA and Apply Memory (LLM prefilling) to GPU (Figure 6(b)). For each segment, the GPU delivers the KV cache to the FPGA and the FPGA returns the token ids for the synthesized memory to the GPU for concatenation.

*Memory as Context*: The kernel mapping follows the General Setup, but with a recurrent loop communication for input segments. We revise data placement as follows: (1) the retrieved memory embedding is transferred to the GPU, which only incurs communication overhead comparable to the retrieved index in the General Setup; (2) memory is stored only on the FPGA, since GPU-side Prepare Memory only requires the retrieved memory and the next input segment to generate new memory. This enables more kernel

fusion on the FPGA for higher efficiency, while sparing GPU memory for model storage.

## 5.3. Kernel Design

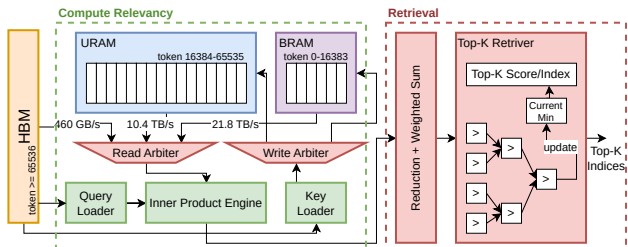

*Figure 7.* Architecture of the FPGA kernel for General Setup. A streaming dataflow design connecting two modules: (a) Compute Relevancy uses fast SRAM (BRAM+URAM) and HBM to store compressed keys, where an inner-product engine consumes queries and computes scores; (b) Retrieval performs partial and weighted sum across query heads and feeds results to a top-$k$ retriever that continuously maintains a running top-$k$ list.

For GPU kernels, we reuse existing optimized libraries, while our design effort focuses on FPGA kernels, which need substantially more development time. For MemAgent and Memory as Context, we adopt the design paradigm of prior FPGA-based LLM accelerators (Yang et al., 2024b; Zeng et al., 2024; He et al., 2025b;c; Zhang et al., 2026a). This section presents the general kernel design. We include more technical detail in Appendix E.

Figure 7 illustrates the FPGA kernel architecture. Here, we present DeepSeek Attention as an example for illustrating the computations. One advantage of FPGAs over GPUs is that users can build streaming dataflow designs: executions are data driven, reducing explicit control overhead and time-consuming off-chip memory accesses. In General Setup, modules are connected in a dataflow manner. The Compute Relevancy module calculates inner product scores between each query head and all key vectors. All past key vectors are stored in a three-level physical memory hierarchy: BRAM with 21.8 TB/s, URAM with 10.4 TB/s, and HBM with 460 GB/s. The BRAM and URAM can hold 40MB of data in total for U55C. Each memory tier has a different capacity, and the key loader writes through the memory hierarchy with a write arbiter. Key vectors with smaller token IDs

are stored in faster memory to maintain high access speed. Vectors are streamed to inner product engine to calculate the score. The resulting scores are streamed to the Retrieval module, which chains a reduction unit and a top-$k$ retriever. The reduction unit aggregates partial sums and computes the final score per key. The top-$k$ retriever maintains a running top-$k$ list by comparing incoming scores using a parallel reduction tree. After scanning all keys, it outputs the indices of the top-$k$ scores.

### 5.4. Deployment

Traversing the methods in Table 1, we implement each step (standalone or fused) as reusable kernels to form a library. Users can further build new algorithm by recombining kernels and interfacing them through our GPU-FPGA communication API (e.g., block-based RAG with BM25 and max-reduction kernels). Arbitrary methods may require custom kernels. We plan to reduce this effort via design automation in our future work.

## 6. Evaluation

### 6.1. Experiment Setup

**Hardware.** For the heterogeneous system, we run experiments on a node with an AMD Alveo U55C FPGA and an AMD Instinct MI210 GPU, with AMD EPYC 7v13 as the host CPU (detail in Appendix D). For the baseline, we use the same GPU model for fair comparison. We do not have physical access to a system with both NVIDIA A100 and U55C on the same node, but we provide estimations based on the real-system profiling in Appendix H to demonstrate the generalizability of our system. Notably, U55C is fabricated in an older process technology than the GPU (16 nm vs. 6 nm) and costs half as much. A newer FPGA model (e.g., AMD Versal V80) can further improve performance.

**Baseline Measurement.** For each workload in Table 1, we use the same datasets as in the original work. For sparse attention, we measure per-token latency under varying past-token lengths; for RAG, MemAgent, and Memory as Context, we measure total request latency (prefill and decode) under different input sequence lengths. We report both end-to-end latency and the fraction attributable to memory processing for the baselines, enabling quantitative evaluation of the acceleration achieved by heterogeneous system. Total latency is measured with Python performance counters, while PyTorch Profiler (Paszke et al., 2019) measures the time fraction spent on memory processing. This is used to derive the latency breakdown without interfered by tracing overhead. To ensure fairness, all methods utilize optimized implementations and identical experimental settings for latency measurement and profiling. Specifically,

- *DeepSeek Attention* (Liu et al., 2025): we use vLLM (Kwon et al., 2023) and modify the codebase to only load the first layer of DeepSeek V3.2 Exp due to the limited GPU memory of MI210.
- *SeerAttention-R* (Gao et al., 2026a): we utilize TileLang (Wang et al., 2026) optimized kernel. The base model is Qwen 3 8B (Yang et al., 2025a) and block size is 64.
- *LServe* (Yang et al., 2025b): We use the LServe codebase and employ HIPIFY (AMD ROCm Software) to port custom CUDA kernels to HIP kernel for the AMD GPU. The base model is Llama 3.1 8B (Grattafiori et al., 2024).
- *RAG*: For single-stage RAG (DRAGIN, FLARE, and Fixed-sentence RAG), we follow the experiment setup in DRAGIN (Su et al., 2024) and utilize Llama 2 7B (Touvron et al., 2023) as the generator model. For two-stage RAG, we adhere the setup of RAG-EDA (Pu et al., 2024) with Llama 3.1 8B as the generator model.
- *Memory as Context*: Since there is no official Titans (Behrouz et al., 2026) implemenetation, we employ the open-source implementation of HMT and update the summarization step into a linear projection to replicate Titans.
- *MemAgent* (Yu et al., 2026): We use Qwen 2.5 7B (Yang et al., 2024a) as the base model and the default hyperparameters defined in the codebase.
- *TTT/LaCT* (Zhang et al., 2026b; Sun et al., 2025): We employ the codebase for LaCT (Zhang et al., 2026b) for profiling and benchmarking.

Details of Hyperparameters in Appendix D.

**Tools.** We utilize ROCm 6.3 for GPU kernel development. FPGA kernels are designed using Xilinx Vitis HLS 2024.2 and implemented with Vivado 2024.2. P2P mode is enabled to allow DMA for the FPGA. Details for cross-device communication is in Appendix C.

### 6.2. Latency

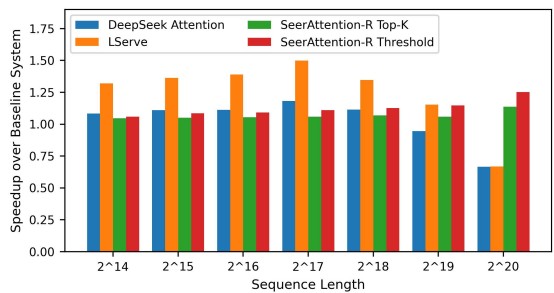

*Figure 8.* End-to-end speedup of the GPU-FPGA heterogeneous system over the baseline for sparse attention mechanisms.

Each method benefits from the GPU-FPGA system in three aspects: 1) the FPGA's large, high-bandwidth on-chip memory permits faster data access than the GPU; 2)

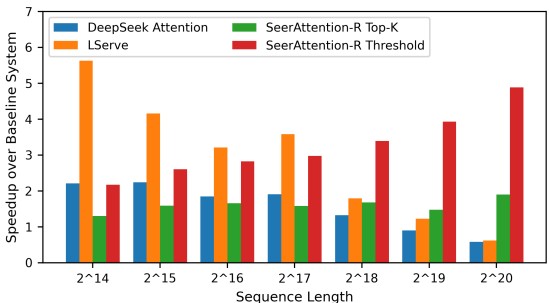

*Figure 9.* Speedup for the memory processing steps deployed on the GPU-FPGA heterogeneous system for sparse attentions.

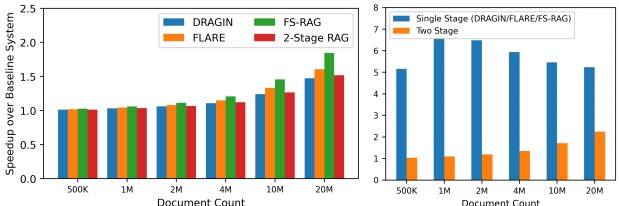

*Figure 10.* Left: End-to-end speedup of the GPU-FPGA system over the baseline for RAG. Right: Speedup of memory processing for the single stage RAG (DRAGIN/FLARE/FS-RAG) and two stage RAG. The reranker of the two stage RAG is executed on GPU.

operations within or across memory processing steps are pipelined through a streaming dataflow, facilitating finer-grained computation-communication overlap than on GPUs; 3) the flexible memory system design maximizes HBM bandwidth utilization, achieving higher decoding throughput even when GPUs offer higher peak bandwidth.

*Case 1: Large On-chip Memory.* By storing compressed key vectors in FPGA on-chip memory (URAM and BRAM), the U55C provides about $5\times$ more effective bandwidth than GPU SRAM for Compute Relevancy and Retrieval. This yields a $1.8$–$2.2\times$ kernel speedup for SeerAttention-R (top-$k$), $2.6$–$4.9\times$ with threshold, and $1.2$–$5.6\times$ for LServe (Figure 9), translating to $1.04$–$1.49\times$ faster end-to-end inference. When the sequence length exceeds 1M tokens, LServe and DeepSeek Attention will experience a drop in speed on the FPGA due to accessing the HBM. Practically, the system can dynamically fall back to GPU-only execution to avoid a performance loss.

*Case 2: Pipelined and Flexible Datapath.* The FPGA exploits fine-grained pipelining and optimized random access to overlap communication and computation in BM25, top-$k$, and dependent memory-processing steps. This benefit generalizes across sparse attention, RAG, and Memory as Context. For DeepSeek Attention, we achieve a $1.3$–$2.2\times$ speedup in memory processing and a $1.1$–$1.2\times$ end-to-end speedup (Figure 9). For RAG, single-stage methods improve memory processing speeds by $5.1$–$6.6\times$ over the BM25S

baseline (Figure 10), while two-stage RAG is limited to $1.1$–$2.1\times$ due to reranker dominance, resulting in up to $1.47$–$1.84\times$ end-to-end speedup. For Memory as Context, fusing query generation with cross attention produces a $3.1$–$4.0\times$ memory-processing speedup and a $1.3$–$1.6\times$ end-to-end speedup (Figure 11).

*Case 3: Faster Decoding.* LLM decoding is fundamentally memory-bound, and FPGA architectures can precisely control HBM transactions to sustain higher effective bandwidth than GPUs, whose peak bandwidth is often underutilized during decoding. Since MemAgent relies on decoding to generate memory, this advantage directly improves system performance. As shown in Figure 12, under prefill–decode disaggregation, the GPU–FPGA system consistently achieves a $1.8\times$ speedup over a GPU-only baseline.

We contain detailed analysis on the source of improvement from the FPGA in Appendix F.

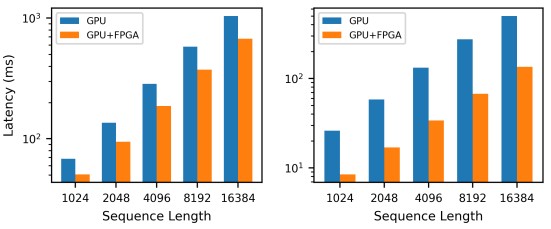

*Figure 11.* Left: End-to-end latency of the GPU-FPGA system vs. the baseline for Memory as Context method. Right: Latency for memory processing in Memory as Context. Similar to Titans, we use a linear projection on the current segment for query generation.

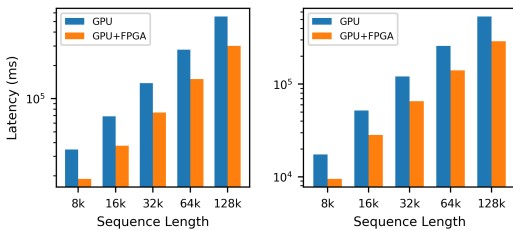

*Figure 12.* Left: End-to-end latency of the GPU-FPGA heterogeneous system vs. the GPU-centric system for MemAgent. Right: Latency for memory processing in MemAgent.

Although the amount of data transferred between the GPU and FPGA ranges from a few indices to long vectors, the resulting communication overhead remains small relative to the end-to-end latency. This is because, under our deployment strategy, methods that require larger data movement also tend to incur substantially higher computation latency. Across all evaluated methods, we observe roughly a three-order-of-magnitude gap between PCIe transfer overhead and end-to-end latency. Detailed breakdowns are provided in Appendix C.1.

*Table 3.* The energy efficiency and the improvement of the GPU-FPGA system over the baseline. DSA stands for DeepSeek Attention, and SA-R denotes for SeerAttention-R.

| Category | Method | GPU-FPGA (J/req. or J/tok) | Baseline (J/req. or J/tok) | Max Improve | Geomean Improve |
|---|---|---|---|---|---|
| Sparse Attn. | DSA | 15.86 | 25.62 | 1.69× | 1.61× |
| | SA-R (Thres.) | 0.30 | 0.34 | 1.26× | 1.14× |
| | SA-R (Top-$k$) | 0.32 | 0.36 | 1.21× | 1.11× |
| | LServe | 0.29 | 0.43 | 1.62× | 1.43× |
| RAG | DRAGIN | 328.16 | 362.57 | 1.34× | 1.10× |
| | FLARE | 241.99 | 275.53 | 1.45× | 1.14× |
| | FS-RAG | 259.88 | 315.25 | 1.71× | 1.21× |
| | Two-stage | 150.33 | 160.68 | 1.23× | 1.07× |
| Synthesized Memory | MemAgent | 3202 | 13662 | 4.66× | 4.66× |
| Memory-as -Context | HMT / Titans | 16.55 | 27.31 | 1.74× | 1.65× |

*Table 4.* Geomean speedup of the GPU-FPGA system over the GPU-centric baseline across batch sizes for various LLM optimization approaches.

| Category | Method | BS=1 | BS=2 | BS=4 | BS=8 | BS=32 |
|---|---|---|---|---|---|---|
| Sparse Attn. | SA-R (Thres.) | 1.12× | 1.21× | 1.33× | 1.47× | 1.60× |
| | DSA | 1.02× | 1.05× | 1.09× | 1.14× | 1.15× |
| | SA-R (Top-$k$) | 1.07× | 1.15× | 1.24× | 1.30× | 1.32× |
| | LServe | 1.19× | 1.34× | 1.51× | 1.67× | 1.83× |
| RAG | DRAGIN | 1.14× | 1.20× | 1.29× | 1.44× | 1.92× |
| | FLARE | 1.19× | 1.26× | 1.38× | 1.55× | 2.11× |
| | FS-RAG | 1.26× | 1.32× | 1.42× | 1.58× | 2.10× |
| | 2-Stage RAG | 1.16× | 1.22× | 1.28× | 1.32× | 1.37× |
| Synthesized Memory | MemAgent | 1.85× | 1.65× | 0.93× | 0.49× | 0.13× |
| Memory-as -Context | HMT/Titans | 1.48× | 1.47× | 1.45× | 1.38× | 1.15× |

## 6.3. Energy Efficiency

Another important aspect of LLM inference is energy efficiency, which directly affects serving cost. Table 3 lists the geomean energy efficiency of the GPU-FPGA system and the baseline for each method mentioned in Section 6.2. Overall, the GPU-FPGA system can shrink the energy cost by $1.11-1.61\times$ for sparse attention, $1.07-1.21\times$ for RAG, $4.66\times$ for MemAgent, and $1.65\times$ for Memory as Context. Furthermore, energy efficiency improvements generally increase with memory size, except for DeepSeek Attention and LServe due to the diminishing performance after 1M tokens for HBM access (stop at $1.43\times$ and $1.07\times$ respectively). The energy reduction does not solely come from the speedup: the FPGA kernels have a lower operating power than the corresponding GPU kernels (Appendix G).

## 6.4. Batch Inference Scaling

Table 4 reports the geomean speedup of the GPU-FPGA system over GPU-centric baselines across different batch sizes. We observe the speedup increases with batch size for sparse attention and RAG methods, decreases for Memory as Context, and slowdown for MemAgent:

• **Sparse attention.** KV cache and latent indexing embeddings are not shared across samples within a batch (Zhao

et al., 2024). Thus, high batch size does not improve data reuse for score computations in sparse attention on GPUs. The GPU-FPGA system can still exploit the advantages of high HBM bandwidth utilization. In contrast, dense components such as linear projections and feedforward layers benefit from weight reuse. As batch size grows, a larger fraction of latency is attributed to sparse attention, amplifying the benefit of FPGA with increasing speedup.

• **RAG.** Methods such as DRAGIN, FLARE, and FS-RAG rely on lexical retrieval with BM25 scoring, where both data access and computation are input-dependent and cannot be shared across batch samples. Similar to sparse attention, batching primarily improves GPU efficiency for dense components but not retrieval. Consequently, the relative cost of retrieval increases with batch size, leading to larger speedups with GPU-FPGA systems. Two-stage RAG can obtain benefits by batch inference through improved weight reuse in the reranker, enhancing GPU utilization and moderating the speedup gain.

• **Memory as Context.** We offloaded the cross attention to the FPGA, which contains linear projections. With larger batch sizes, these linear projections achieve higher weight reuse and improved GPU utilization. This cuts the relative advantage of offloading, resulting in decreasing speedup. However, since the memory embeddings remain independent across samples, FPGA acceleration still provides benefits in long-sequence regimes for computing the cross attention score and performing memory embedding selections.

• **MemAgent.** The memory processing pipeline in MemAgent is a standard LLM inference. Under batching, the decode stage significantly benefits from weight reuse on GPUs. Given the lower compute throughput of FPGAs for dense operations, this leads to performance degradation as batch size increases.

For MemAgent, the system can dynamically select the optimal configuration. For example, when the batch size is larger than 2 in MemAgent, we switch to a GPU-centric deployment to avoid slowdown.

## 7. Conclusion

In this work, we demonstrate that a GPU-FPGA heterogeneous system can accelerate memory processing in LLM inference. By unifying LLM inference optimizations into a four-step memory processing pipeline, we expose its nontrivial contribution to end-to-end latency and exploit its computational heterogeneity with a GPU-FPGA design, achieving a $1.04\sim2.2\times$ speedup and $1.11\sim4.7\times$ energy reduction. While our prototype uses off-the-shelf devices, a heterogeneous ASIC could further improve energy efficiency and eliminate cross-device communication overhead.

## Impact Statement

This paper presents work whose goal is to advance the field of efficient machine learning inference. There are many potential societal consequences of our work, none of which we feel must be specifically highlighted here.

## Acknowledgement

This work was supported in part by PRISM (2023-JU-3135), one of the seven centers in the JUMP 2.0 program sponsored by SRC and DARPA. It is also supported by CDSC industrial partners and the AMD HACC Program.

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

# A. Related Works

**Surveys on memory in LLM inference.** Recent surveys on LLM memory provide comprehensive characterizations of memory types and their usage patterns. Wu et al. draw parallels between LLM memory and human cognition, proposing a three-dimensional, eight-quadrant (3D-8Q) taxonomy based on memory origin, representation form, and retention duration. Zhang et al. evaluate memory effectiveness across four categories: parametric memory (model weights), contextual memory (KV caches), external memory (indexed vectors), and procedural memory (event stores). However, these surveys do not articulate the shared procedural structure by which memory is generated, accessed, and updated. This omission hinders a systematic understanding of memory processing in LLM inference and limits opportunities for optimizations.

**Sparse attention.** Longformer (Beltagy et al., 2020) first proposes an effective sparse attention that combines the sliding window attention and a global token attention. Reformer (Kitaev et al., 2020) employs local sensitive hashing to select the relevant tokens to compute the attention scores. MInference (Jiang et al., 2024) employs mixed static-dynamic sparsity across heads for high precision sparse attention at the prefill stage. SeerAttention (Gao et al., 2026b) trains an auxiliary predictor to identify important blocks. These methods primarily optimize input processing rather than decoding.

**Advanced RAG systems.** JudgeRank (Niu et al., 2024) leverages prompt engineering to distill key information from each document, utilizing users' queries before passing to the reranker. HLATR (Zhang et al., 2022) adds a second reranking pass: it concatenates all candidate documents selected during the initial retrieval stages and then computes multiple similarity scores over this combined text using the second reranker model. By scoring the concatenated content all at once, HLATR provides more holistic relevance judgments.

**Context compression and recurrent models.** QwenLong-L1.5 (Shen et al., 2025) extends MemAgent by introducing planning tokens for structured memory generation. RMT (Bulatov et al., 2022) suggests a recurrent memory transformer that compresses sequences into embeddings but lacks explicit retrieval mechanisms. Mamba (Gu & Dao, 2024) introduces a selective state space model that enables efficient long-context processing by replacing attention with recurrent state updates. Gated DeltaNet (Yang et al., 2025c) improves Mamba by controlling state updates through learnable gates. RWKV (Peng et al., 2023) unifies RNN and Transformer paradigms by utilizing time-mixing and channel-mixing mechanisms, enabling constant-memory inference.

# B. Detail Computation Properties of Memory Processing Pipeline

Figures 13 and 14 illustrates the arithmetic intensity of memory processing in each LLM inference optimization. Higher arithmetic intensity means the operation is more compute bounded. The arithmetic intensity and computation pattern analysis are widely discussed in prior works (Chen et al., 2024) related to LLM inference acceleration. For the latency distribution among each step of memory processing pipeline:

- *Sparse Attention & RAG*: the bottleneck is compute relevancy and retrieval, and the proportion of latency is increasing as the memory size grows.

- *MemAgent*: the bottleneck is prepare memory, which is essentially LLM decoding.

- *Memory as Context*: similar to sparse attention and RAG, the bottleneck is compute relevancy and retrieval, but the proportion of latency grows more slowly than them.

- *TTT/LaCT*: the bottleneck is prepare memory and apply to inference, which is the LaCT block forward and backward pass.

For sparse attention, RAG, MemAgent, and Memory as Context methods, the dominant latency originates from memory-centric and irregular operations such as relevance computation and top-$k$ retrieval. By offloading these bottleneck steps to the FPGA, which enables customized data paths and fine-grained control, the system mitigates GPU inefficiencies on memory-bounded workloads and shortens the critical path of inference. Consequently, this mapping yields a substantial improvement in end-to-end inference speed.

# C. Cross-device Communication

For generalization across different vendors of FPGAs and GPUs, we consider PCI Express (PCIe) as the interconnect standard to communicate between devices. The devices are installed in the same node with the same root complex to

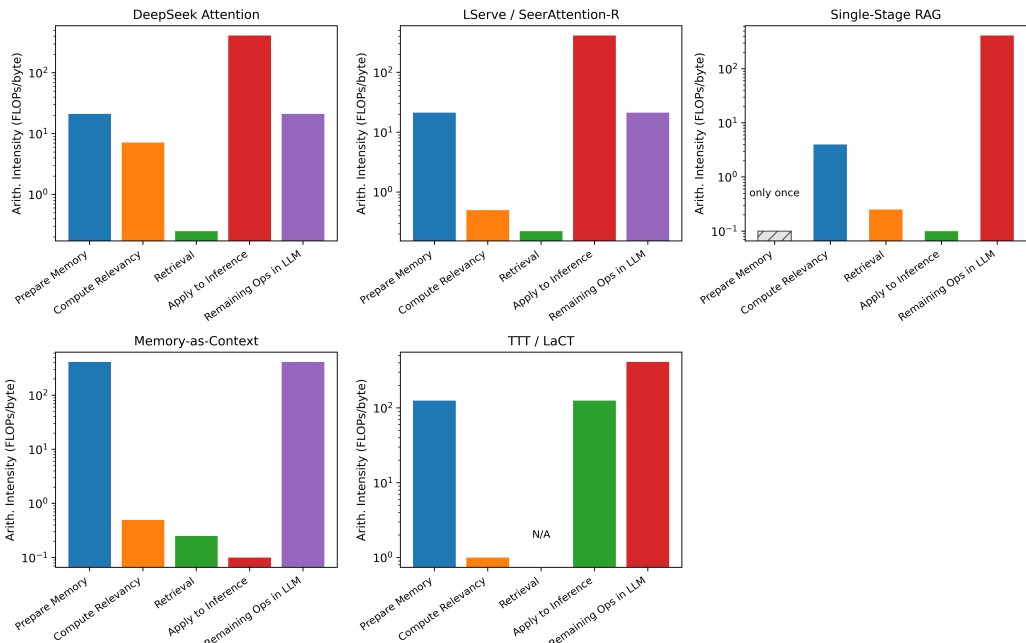

*Figure 13.* Arithmetic intensity (FLOPs/byte) of memory processing pipeline and the rest operations in LLM inference for sparse attention, single-stage RAG, Memory as Context, and TTT/LaCT. For RAG, prepare memory is a one-time operation and will be amortized with multiple queries.

minimize PCIe overhead. The standard method of communicating FPGA with GPU is through memory copy runtime APIs of each device and utilize CPU and system DRAM to handle the control (Yang et al., 2024b) (Figure 15(a)). This method can only achieve 1/20 of the peak PCIe bandwidth. In this work, we configure the system alternatively using PCIe peer-to-peer (P2P) data transfer: Both FPGA's and GPU's HBM can initiate direct memory access (DMA) to the pinned memory buffer (non-swappable by the operating system) allocated on CPU (Figure 15(b)). The transaction will involve two hops of DMA with CPU as the intermediate, bypassing the system DRAM.

A common concern regarding PCIe-based data transfer is its limited bandwidth. Compared to GPU–GPU communication over NVLink (600 GB/s for NVLink 3.0), PCIe provides substantially lower throughput (32 GB/s for PCIe 3.0). However, our profiling results indicate that the communication overhead introduced by PCIe in our configuration is sufficiently low and more than compensated for by the performance gains achieved through our customized FPGA kernels (Section 6.2).

### C.1. PCIe Latency

One concern of the GPU-FPGA system in accelerating memory processing in LLM inference is the PCIe overhead between the FPGA and the GPU. Figure 16 depicts the latency profiling of PCIe latency. For sparse attention and RAG, the data transferred between the FPGA and the GPU are retrieved index and processed memory, which are under 1KB per request. Consequently, the transaction latency is in the order of microseconds (7 $\mu$s). For MemAgent and Memory as Context, the transferred embeddings have a size in the order of MBs, which takes microseconds to miliseconds to deliver. PCIe overhead is not the bottleneck because methods that require larger data movement also exhibit substantially longer end-to-end latency, making the PCIe overhead relatively small in comparison. The following numbers provide an order-of-magnitude comparison between PCIe latency and corresponding GPU kernel latency:

- **Sparse Attention:** Transfers include the query vector, new key indexing vectors, and retrieved indices, taking approximately 14 $\mu$s. The corresponding GPU kernels take 128–2450 $\mu$s.

- **RAG:** Transfers only include the query document counter and retrieved indices, taking approximately 7 $\mu$s. The corresponding GPU kernels take 23–1596 ms.

- **Memory as Context:** Transfers include memory, query, and retrieved embeddings for each segment, taking approximately 20–320 $\mu$s. The corresponding GPU kernels take 26–498 ms.

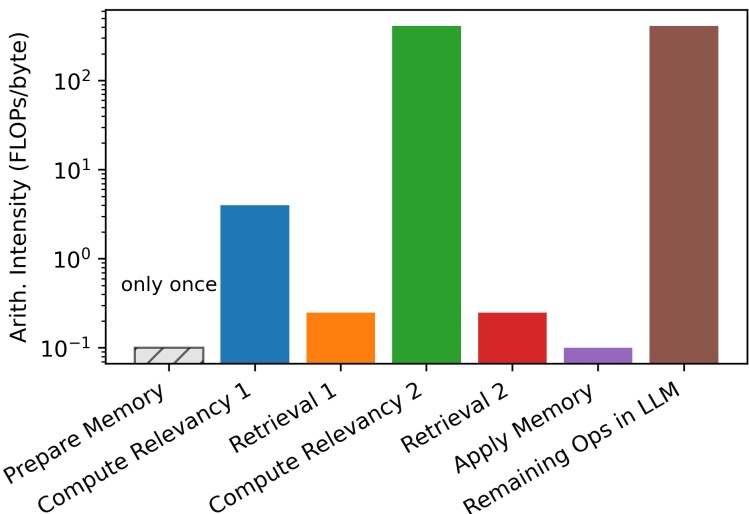

*Figure 14.* Arithmetic intensity of memory processing pipeline and the rest operations in LLM inference for two-stage RAG. Each stage has a compute relevancy and retrieval step.

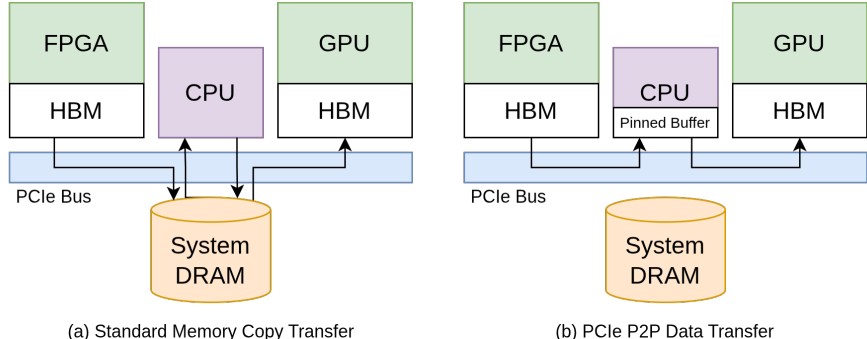

*Figure 15.* (a) Standard cross device data transfer using memory copy for heterogeneous system. (b) PCIe P2P data transfer to bypass system DRAM accesses.

- **MemAgent:** Transfers include KV cache and token IDs for each segment, taking approximately 14–218 ms. The corresponding GPU kernels require 17–534 s.

These comparisons show that PCIe communication overhead remains small ($\sim 1000\times$ difference) relative to computation time across all methods, even when data transfer size increases.

## D. Detail Experiment Settings

The system specification is shown in Table 5.

*Table 5.* Hardware Specification

| Device | Model | Memory |
| --- | --- | --- |
| CPU | AMD EPYC 7v13 @ 2.5GHz (7 nm) | 1 TB (51.2 GB/s) |
| GPU | AMD Instinct MI210 (6 nm) | 64 GB (1.6 TB/s) |
| FPGA | AMD Alveo U55C (16 nm) | 16 GB (460 GB/s) |

**DeepSeek Attention** DeepSeek Attention (Liu et al., 2025) introduces a lightning indexer to the Multi-headed Latent Attention (MLA) (Liu et al., 2024) module. For every compressed query embedding and KV latent embedding, it first generates 64 query heads and a key indexing vector by applying partial RoPE embedding, computes the dot products

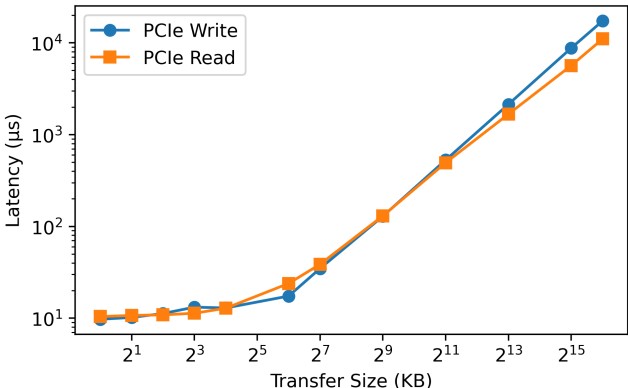

*Figure 16.* Transfer latency on PCIe bus against the transfer data size. For transferring indices (KB) and single-token KV embeddings (MB) are in the order of microseconds, which is negligible compared to the latency of memory processing.

between key vectors and all query heads, and weight-averages them based on the query weights derived from the input token. The final scores are used for top-$k$ selection for MLA module to attend the token, where $k$ is 2048 in the DeepSeek V3.2 Exp model.

We utilize the vLLM (Kwon et al., 2023) optimized DeepSeek V3.2 Exp inference kernel for the GPU baseline. Since the original DeepSeek V3.2 Exp is larger than the HBM capacity of MI210 GPU, we modify vLLM to load only the first layer of the model. Since $k$ of DeepSeek attention is constant across layers, we calculate the end-to-end latency by multiplying the first-layer latency with the number of layers. For the GPU-FPGA system, the multi-head query vector and new key indexing vectors are delivered to FPGA from GPU and the retrieved indices are transferred from FPGA to GPU.

**SeerAttention-R** During inference, SeerAttention-R (Gao et al., 2026a) first down projects the query vectors and key vectors with average pooling, then computes the dot products between the the projected queries and keys. Each score is for a block of tokens. The attention module only attend the current tokens to the selected blocks. During inference, the user can choose to determine the retrieved tokens by either a token budget with top-$k$ selection, or a threshold based selection (select if the score is above the threshold).

We utilize the TileLang (Wang et al., 2026) optimized kernel for SeerAttention-R. The base model is Qwen 3 8B. We set the block size to 64, token budget to 4096 for top-$k$ mode, and threshold to 5e-4 for threshold mode. For the GPU-FPGA system, the projected query vector and new block-wise key indexing vectors are delivered to FPGA from GPU and the retrieved indices are transferred from FPGA to GPU.

**LServe** LServe (Yang et al., 2025b) organizes blocks of tokens into logical pages and physical pages, where each physical page can have multiple logical pages. Each logical page is represented by two vectors: one with the minimum and one with maximum values for each channel. The score is computed by finding the maximum dot products between the query and these two vectors. Selection is in the granularity of the physical page, where the score for each physical page is the maximum score of logical pages.

To use the CUDA kernels in the LServe codebase, we use HIPIFY (AMD ROCm Software) to port them to HIP kernel and compile them with `hipcc` to deploy on the MI210 GPU. We employ the default hyperparameters in their original benchmarking script for profiling. For the GPU-FPGA system, the query vector and min-max indexing vectors are delivered to FPGA from GPU and the retrieved indices are transferred from FPGA to GPU.

**DRAGIN, FLARE, and Fixed-sentence RAG** Following the experiment setup in DRAGIN (Su et al., 2024), all three methods utilize Llama 2 7B (Touvron et al., 2023) as the generator model and BM25 indexing as the retrieval heuristic. The original retriever backend is ElasticSearch (Elasticsearch, 2026). We replace it with a faster backend specific for BM25 indexing (BM25S (Lù, 2024)). The system will retrieve 64 documents and the maximum number of tokens generated is 32. For the GPU-FPGA system, query sentences are preprocessed into word counts and passed to the FPGA as a dictionary, and the FPGA return the retrieved document indices back to the GPU.

**Two-stage RAG** A two-stage RAG first executes an hybrid search (semantic embedding and BM25 lexical search) to retrieve the top-$N$ relevant documents, then filters the documents using a reranker to obtain the top-$k$ documents in the

selected $N$ documents. The reranker is usually a transformer model. In the experiment, we follow the setup in RAG-EDA (Pu et al., 2024), where the first stage comprises an embedding model (`bge-large-en-v1.5` (Xiao et al., 2024)) and a BM25 indexer, and the second stage is a reranker (`bge-reranker-large` (Xiao et al., 2024)). The first stage selects 64 documents and the second stage picks 10 documents. The maximum number of tokens generated is 32.

**Memory as context** In Titans (Behrouz et al., 2026), Memory as Context is a type of recurrent models that chunks sequence into segments, utilize soft prompts to generate latent embeddings as memory, and convert each segment into query embedding to find relevant embeddings stored in the past. In the experiment, we set the segment length to 1024 tokens and the output sequence length is 32. In the GPU-FPGA system, the GPU delivers the newly generated memory and next segment embeddings to the FPGA, and the FPGA returns the retrieved memory embeddings to the GPUs.

**MemAgent** Following the experiment setup in Yu et al., we set the segment length to 5000 tokens, the memory size to 1024 tokens, and the output max token length to 32. In the GPU-FPGA system, the GPU sends the KV cache produced in the prefill stage to the FPGA, and the FPGA returns the generated memory token ids back to the GPUs.

## E. FPGA Kernel Design

For GPU kernels, we use (cuBLAS/cuSparse (NVIDIA Corporation) and rocBLAS/rocSparse (Advanced Micro Devices, Inc.)) for linear operations and custom CUDA/HIP kernels for non-linear operations to ensure that steps deployed on the GPU achieve state-of-the-art performance.

For Memory as Context, we follow the HMT plugin design in FlexLLM (Zhang et al., 2026a). The FPGA loads the segment embeddings to the HBM from the CPU when the input is streaming. As show in Figure 17, the segment loader will read the segment embeddings on chip and stream to the query linear projection module to generate the query vector. At the same time, the memory loader reads past generated memory embeddings and performs cross attention with the query to extract and amplify the memory that are most relevant to the current segment. The output memory will be written back to the HBM and delivered to the GPU for processing the proceeding segments. Each modules is connected with FIFO streams.

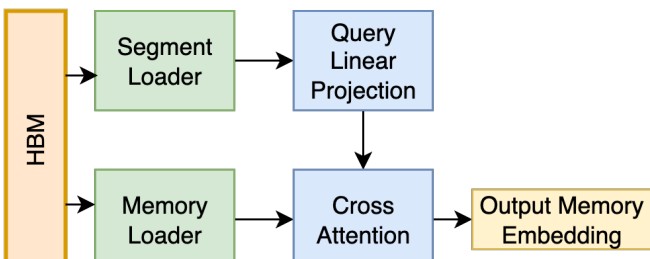

*Figure 17.* FPGA kernel architecture for Memory as Context method. The kernel is a dataflow design with each module fully data driven compute a single operation. Past memory embeddings are cached in the HBM and the segment embeddings are loaded from the CPU directly for each incoming segment.

For MemAgent, the FPGA only executes the LLM decoding. Therefore, we can directly follow the designs in previous works (Zeng et al., 2024; He et al., 2025c; Yang et al., 2024b) and optimize them to specialize for decoding, e.g., the attention is a sequence of GEMV operations and we can increase the parallelism in the hidden dimension. Figure 18 illustrates the overall architecture. We follow a similar design to FlightLLM (Zeng et al., 2024) with separate special function units (SwiGLU, LayerNorm) and matrix/vector multiplication engines (Linear Projection, Attention). However, we align with the LUT-LLM (He et al., 2025c) with separate attention and linear projection engines since attention requires higher precision than linear projections to maintain accuracy. A global buffer is used to store partial weight matrices and intermediate data. Data are streamed in each engine to reduce on-chip data store requirements, and executions are sequentially scheduled between engines to ensure high computational throughput.

## F. FPGA Kernel Improvement Analysis

*Case 1: Large On-chip Memory.* For block-sparse attention, compressed key vectors are either fully stored (SeerAttention-R) or partially cached (LServe) in FPGA on-chip memory. The aggregated on-chip memory (URAM + BRAM) bandwidth on U55C is about $5\times$ higher than the effective SRAM bandwidth of MI210, with each bank managed as a scratchpad

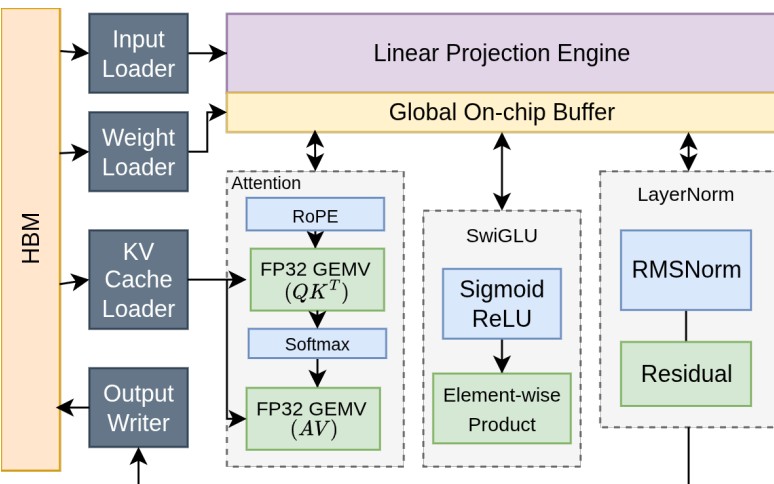

*Figure 18.* FPGA kernel architecture for MemAgent. Following the design in prior works (Zeng et al., 2024; He et al., 2025c;b), we design the kernel specialized to LLM decoding, where the KV cache is delivered from the GPU through PCIe. Linear projections are executed in INT4 to align with the weight precision, and the rest operations (attention, SwiGLU, LayerNorm) are calculated at FP32 to maintain accuracy of the model.

to maximize target data access bandwidth. Since the Compute Relevancy and Retrieval stages are memory-bound, this yields 1.8–2.2× speedup for SeerAttention-R with top-$k$, 2.6–4.9× with threshold, and 1.2–5.6× for LServe (Figure 9), translating to 1.04–1.25× and 1.15–1.49× end-to-end speedups for SeerAttention-R and LServe. LServe performance degrades beyond 256K tokens and becomes worse than the GPU at 1M tokens. This is because the FPGA kernel starts to read from the HBM, while the GPU's HBM utilization scales with sequence length and eventually surpasses that of the FPGA at 1M tokens. Therefore, the system **dynamically falls back** to GPU-only execution for sequences longer than 1M tokens.

*Case 2: Pipelined and Flexible Datapath.* The memory processing pipeline has strong data dependency between steps. Moreover, operations such as the BM25 and the top-$k$ selection involve irregular data access pattern as illustrated in Section 4. With custom logic for fine-grained pipelining and optimized random access, the U55C FPGA can better reduce and overlap communication and computation latency than GPUs. Some methods (e.g., RAG) adopt CPU offloading as a baseline to accelerate these operations relative to GPU execution. Nevertheless, the U55C still achieves higher performance due to its 3.5× higher peak TOPs and its substantially higher HBM bandwidth compared to system DRAM.

We observe that these benefits generalize across sparse attention, RAG, and Memory as Context methods. For sparse attention, we pipeline index score computation with top-$k$/threshold-based selection, achieving a 1.3–2.2× speedup in memory processing for DeepSeek Attention, resulting in a 1.1–1.2× end-to-end speedup. Note that both LServe and DeepSeek attention require reading key vectors from HBM for a subset of tokens, incurring lower bandwidth than on-chip memory. Similar to LServe, the system dynamically falls back to GPU-only execution when the sequence length exceeds 1M. For RAG (Figure 10), we pipeline the BM25 score computation with embedding search and top-$k$ selection. For single-stage RAG methods (DRAGIN, FLARE, and FS-RAG), the GPU-FPGA system achieves a 5.1–6.6× speedup in memory processing over the baseline with the state-of-the-art BM25S kernel. In contrast, for two-stage RAG, the speedup drops to 1.1–2.1× due to the reranker dominating the GPU execution time, leading to an overall end-to-end speedup of up to 1.47–1.84×. For Memory as Context methods, we fuse query generation, which is a linear projection on the current segment embeddings, with cross attention that outputs memory embeddings. Figure 11 compares end-to-end and memory-processing latency for Memory as Context between the GPU-FPGA system and the baseline. The GPU-FPGA system achieves a 3.1 − 4.0× speedup for memory processing, resulting in a 1.3 − 1.6× end-to-end speedup.

*Case 3: Faster Decoding.* Prior studies have shown that FPGAs can achieve faster LLM decoding than GPUs despite having fewer raw computational resources (Zeng et al., 2024; He et al., 2025b;c). This advantage arises because LLM decoding is fundamentally memory-bound, and FPGAs enable customized architectures that precisely control off-chip memory transactions to fully exploit the HBM bandwidth. In contrast, although GPUs provide higher peak HBM bandwidth, this bandwidth is often underutilized even with highly optimized kernels (Zeng et al., 2024). Since MemAgent relies on

LLM decoding to generate memory, the decoding efficiency of FPGAs directly translates into substantial system-level benefits. As shown in Figure 12, under the prefill-decode disaggregation scheme, the GPU-FPGA heterogeneous system consistently achieves a $1.8\times$ speedup over a GPU-only baseline.

## G. Expanded Results for Sparse Attention and RAG

Figures 19, 20, 21, and 22 illustrate the absolute value of latency for end-to-end inference and memory processing of sparse attention and RAG. Figures 23 and 24 show the corresponding energy efficiency comparison in joule per token (for sparse attention) and joule per request (for RAG).

For kernel power consumption:

- DeepSeek Attention: U55C: 26.4 W, MI210: 55 W

- SeerAttention-R threshold: U55C: 24.9 W, MI210: 45 W

- SeerAttention-R top-$k$: U55C: 25.3 W, MI210: 46 W

- LServe: U55C: 26.2 W, MI210: 47 W

- RAG: U55C 29.7 W, MI210: 106 W, EPYC 7v13: 34 W

- MemAgent: U55C: 44.2 W, MI210: 99 W

- HMT/Titans: U55C: 42.6 W, MI210: 94 W

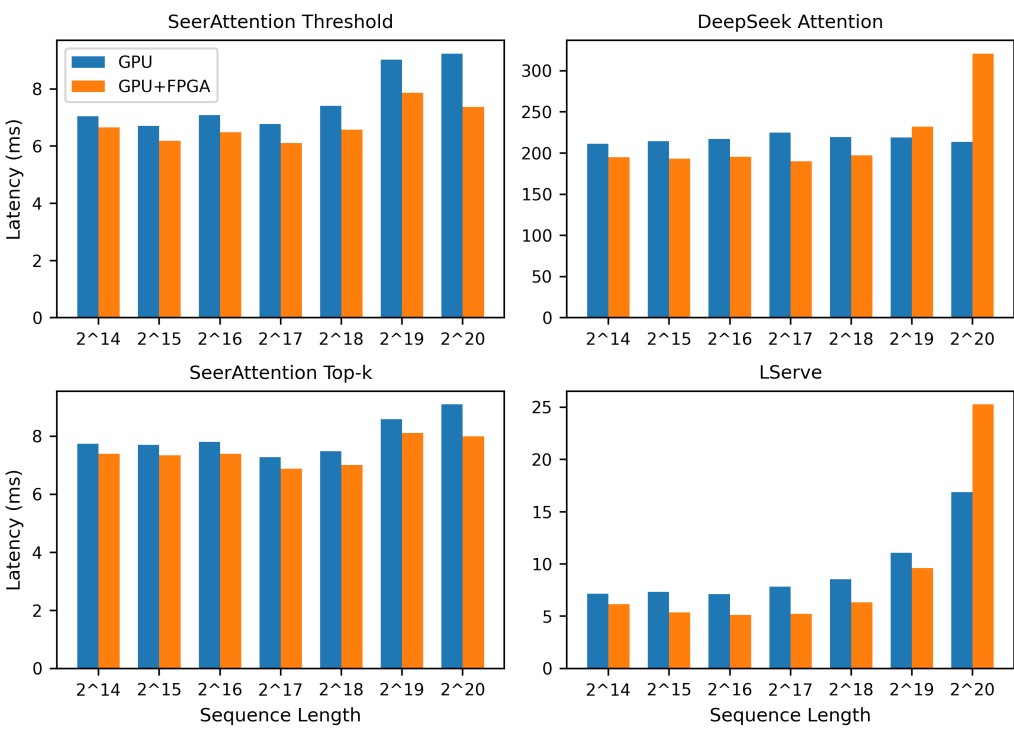

*Figure 19.* End-to-end latency of each sparse attention mechanism on the baseline and the GPU-FPGA system with respect to the sequence length.

## H. Results with NVIDIA A100

Compared with the AMD MI210, the NVIDIA A100 is a more widely adopted GPU for LLM inference. Although we do not have access to a platform that hosts both the A100 and the U55C within the same node, we calculate the end-to-end latency

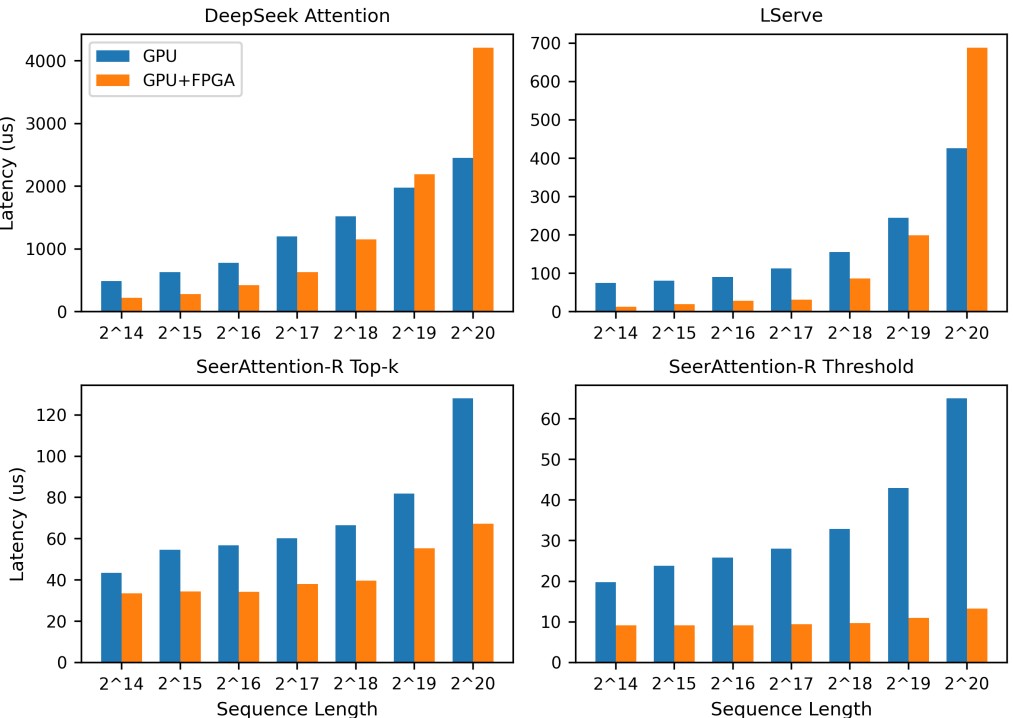

*Figure 20.* Latency of memory processing in each sparse attention mechanism with respect to the sequence length.

by aggregating the measured latency components of the FPGA, GPU, and PCIe communication, while profiling kernel execution latency separately. Figures 25 and 26 present a representative case study using DeepSeek Attention. The results show that, even when paired with the MI210, the GPU–FPGA heterogeneous system can outperform the A100 in certain configurations. For end-to-end latency, since the A100 generally outperforms the MI210 under identical optimizations, the MI210+U55C configuration can be slower than an A100-only system. However, when the GPU is upgraded to the A100, the heterogeneous system continues to deliver consistent speedups. This demonstrates that the proposed heterogeneous system is effective and largely agnostic to the specific GPU model.

## I. Practical Impact and Device Scaling

We add the following clarifications to better contextualize the practical impact of the reported speedups. Our current prototype utilizes the AMD Alveo U55C FPGA, which is manufactured in a 16 nm process technology, while the A100 GPU in our baseline system uses a 7 nm process. More recent FPGA platforms, such as the AMD Alveo V80, provide a substantially higher compute resources and HBM bandwidth. Based on our kernel synthesis results, we estimate that migrating from U55C to V80 would provide an additional $1.6\times$ geomean end-to-end speedup. Since the kernel design paradigms remain unchanged, this migration is straightforward from an implementation perspective.

Although we do not have access to a current-generation GPU-FPGA system for real-system profiling, the architectural trends suggest that the advantage of heterogeneous execution will continue to hold. As GPUs improve, CPU-offloaded or otherwise non-GPU components can become a larger fraction of end-to-end latency, increasing the impact of the FPGA acceleration. Moreover, many targeted operations are memory-bound or irregular, and their performance does not scale linearly with the GPU HBM bandwidth alone. For example, a radix-based top-$k$ on GPUs involves irregular memory accesses for histogram construction and is also constrained by scratchpad capacity, which determines bucket sizing and parallelization. Modern FPGA platforms such as V80 provide 94 MB on-chip scratchpad, which is substantially larger than GPU L1 scratchpad capacity, enabling more efficient implementations of such irregular memory-processing operations.

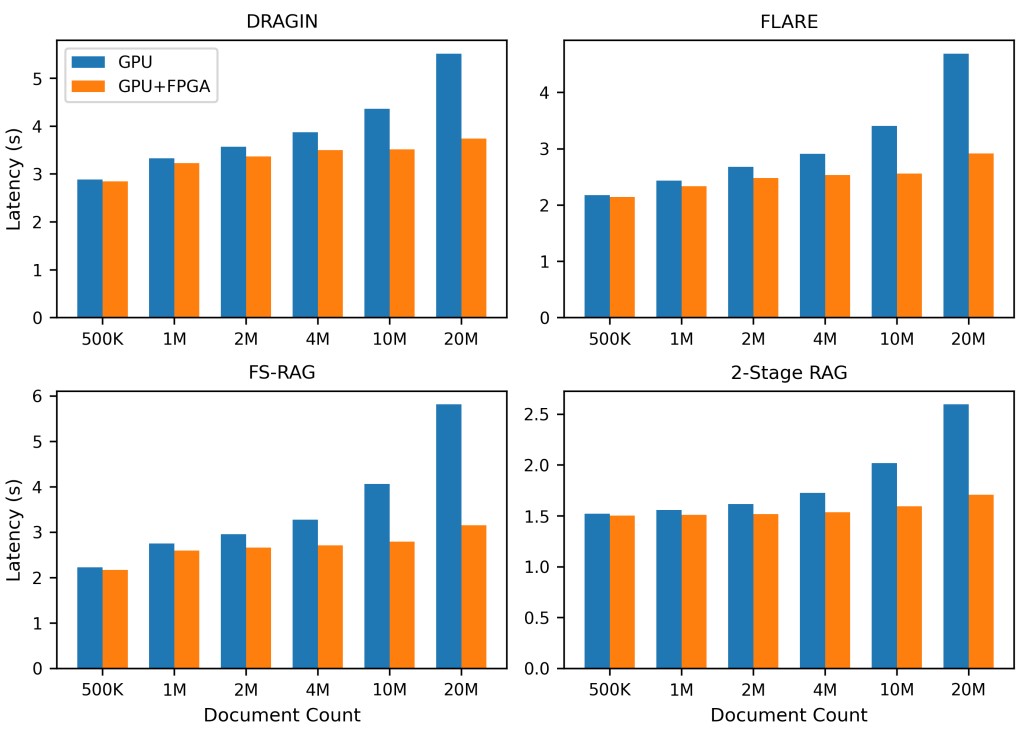

*Figure 21.* End-to-end latency of each RAG system on the baseline system and GPU-FPGA system with respect to the document counts.

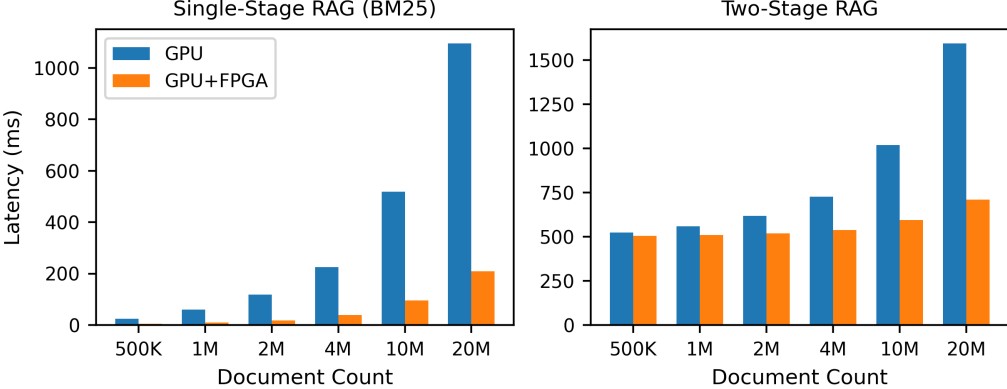

*Figure 22.* Latency of memory processing in single stage RAG using BM25 as the retrieval heuristic (DRAGIN, FLARE, FS-RAG) and two stage RAG with respect to the document counts.

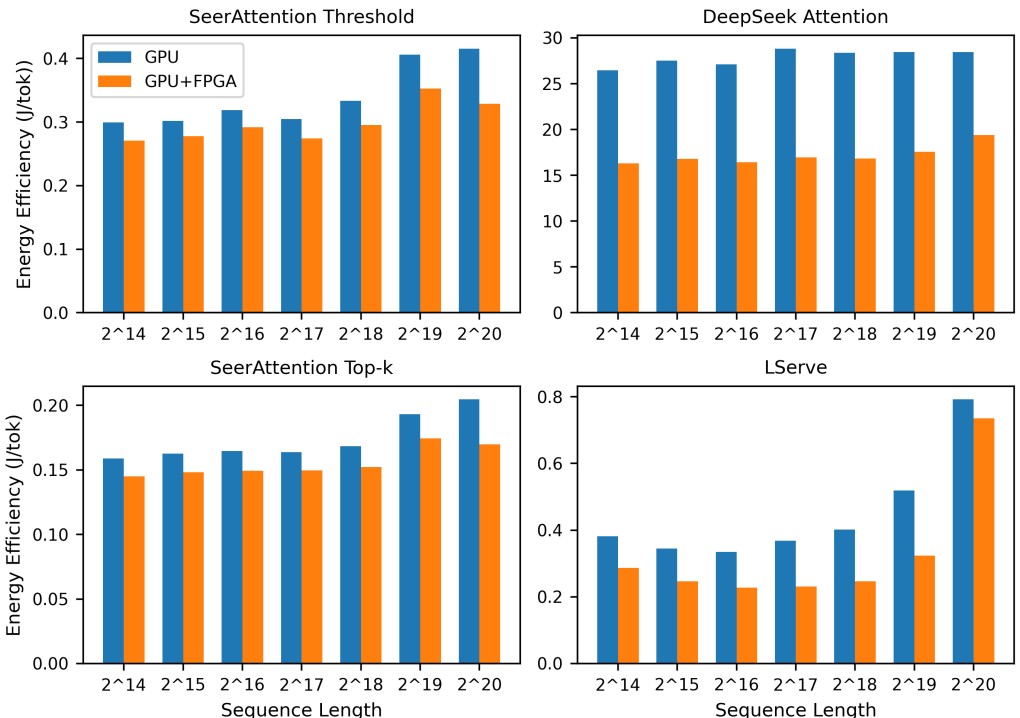

*Figure 23.* Energy efficiency of sparse attention mechanisms in Joule per token.

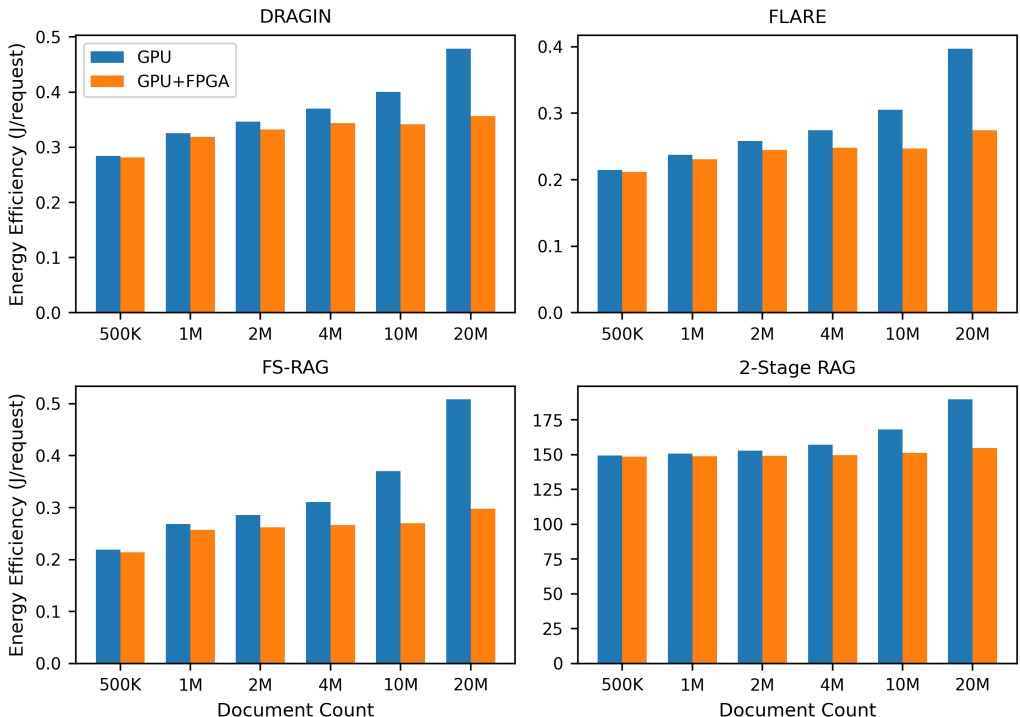

*Figure 24.* Energy efficiency of RAG systems in Joule per request.

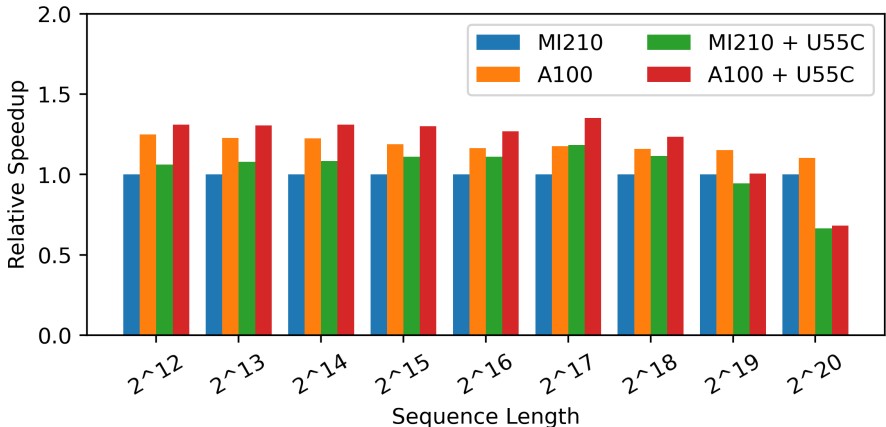

*Figure 25.* The relative speedup of end-to-end inference of DeepSeek V3.2 Exp with DeepSeek Attention when deployed on MI210, A100, MI210 + U55C, and A100 + U55C. A100 is generally faster in LLM inference than MI210. When integrating U55C with A100, the GPU-FPGA heterogeneous system can still speed up the inference.

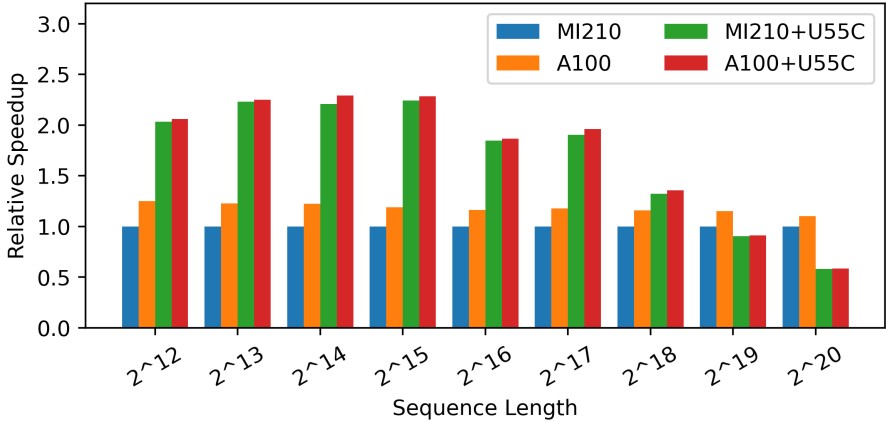

*Figure 26.* The relative speedup of memory processing in DeepSeek Attention when deployed on MI210, A100, MI210 + U55C, and A100 + U55C. Even with MI210, the GPU-FPGA heterogeneous system can still outperform A100.

