# OpenReview forum: "Understand and Accelerate Memory Processing Pipeline for Large Language Model Inference"
_ICML.cc/2026/Conference — ICML 2026 regular_

### Official Review · Reviewer_eQks · 2026-03-11

**Soundness:** 3
**Presentation:** 3
**Significance:** 3
**Originality:** 3
**Overall Recommendation:** 4
**Confidence:** 3

**Summary:**

This paper pertains to unifying diverse memory optimizations (such as sparse attention, RAG, and compressed memory) into a single four-step pipeline, and accelerating it using a co-designed GPU-FPGA heterogeneous architecture. By offloading irregular, memory-bound tasks to the FPGA while retaining dense, compute-bound tasks on the GPU, the proposed system exploits the specific hardware strengths of each device to achieve up to 2.2× latency speedups and 4.7× energy reductions across multiple state-of-the-art models.

**Compliance With Llm Reviewing Policy:**

Affirmed.

**Final Justification:**

The rebuttal addressed most of my concerns.

**Key Questions For Authors:**

1. How would the proposed FPGA advantages hold up against current-generation, high-bandwidth GPUs (e.g., NVIDIA H100 or AMD MI300X)? At what point does the massively scaled HBM bandwidth of newer GPUs negate the benefits of offloading to an FPGA?
2. How does the GPU-FPGA heterogeneous system perform under large batch sizes? Does the FPGA's limited compute capacity become a bottleneck when processing relevancy scores for hundreds of concurrent queries, or does the streaming architecture effectively mask this?
3. What are the specific architectural or algorithmic reasons for the high variance in speedups across the different sparse attention mechanisms shown in Figures 8 and 9?
4. Could you provide a detailed, quantitative timeline or breakdown showing the exact PCIe transfer latency versus the FPGA kernel execution time?

**Limitations:**

Same as in weaknesses.

**Strengths And Weaknesses:**

**Strengths:**

1. **Unified Abstraction:** The proposed four-step memory processing pipeline is highly general and synthesizes disparate techniques (e.g., DeepSeek MLA, RAG, MemAgent, Titans/HMT) into a single, cohesive framework.
2. **Effective Heterogeneous Mapping:** The design smartly leverages the strengths of both devices—using the FPGA's streaming dataflow, high on-chip memory bandwidth, and bit-level control for operations like TopK selection and BM25, while leaving dense linear algebra to the GPU.
3. **Strong Empirical Results:** The evaluation spans a diverse set of SOTA optimization methods. The latency and energy improvements (especially the 4.66× energy reduction for synthesized memory like MemAgent) are substantial and practically meaningful for LLM serving.

**Weaknesses:**

1. **Hardware Platform Generalization and Baseline Age:** The physical evaluation is entirely restricted to a homogeneous AMD environment (MI210 GPU + U55C FPGA). While Appendix H offers theoretical latency estimations for an NVIDIA A100 setup, the absence of physical cross-vendor testing leaves critical questions unresolved. Furthermore, the baseline GPU (MI210) is a previous-generation architecture. Evaluating the system against more contemporary, high-bandwidth GPUs (e.g., NVIDIA H100 or AMD MI300X) is necessary to prove that the FPGA's advantages on memory-bound workloads still hold as GPU HBM capabilities drastically scale.
2. **Absence of High-Throughput and Batching Analysis:** The latency numbers and experimental setup predominantly reflect single-request or low-batch-size scenarios. Because production LLM serving environments rely heavily on large concurrent batching to maximize system throughput, this omission is significant. It remains unclear whether the FPGA's streaming dataflow and limited on-chip compute capacity can effectively scale under heavy concurrent batching, or if the FPGA would bottleneck the GPU during high-throughput workloads.
3. **Insufficient Granularity in Performance Analysis:** The manuscript lacks a deep dive into the variance and underlying bottlenecks of its own results. For instance, Figures 8 and 9 demonstrate highly variable speedups across different sparse attention mechanisms, yet the text does not thoroughly analyze the architectural or algorithmic reasons for these discrepancies. Additionally, the authors claim in Appendix C that PCIe communication overhead is "sufficiently low and more than compensated for by the performance gains," but the evaluation fails to provide a concrete, quantitative breakdown of this overlap. Explicitly profiling and presenting the PCIe transfer latency versus kernel execution time is needed to substantiate this claim.

---

> ### Author Rebuttal · Authors · 2026-03-29
>
> Thank you for the feedback. Our responses are the following.
>
> > offers theoretical latency estimations for an NVIDIA A100 setup
>
> Our estimation for A100+U55C is based on real-system profiling for GPU and FPGA separately (See Q1 of review eaft for details).
>
> > How would the proposed FPGA advantages hold up against current-generation, high-bandwidth GPUs
>
> For computations that the SoTA baselines have already offloaded to CPU (e.g., BM25 Score in RAG), FPGA will still have advantage. For other cases, if we upgrade the FPGA from U55C to the latest V80, the overall advantage will still hold. Although we do not have access to a system with current-generation GPU + FPGA for real-system profiling, we would like to provide some insights:
>
> - For cases where CPU offloading is involved, upgrading GPU will increase the latency contribution of the non-GPU portion, increasing the impact of FPGA acceleration to the end-to-end latency.
> - For memory-bound workloads, speedup does not scale linearly with HBM bandwidth. For example, radix-based top-k on GPUs involves irregular memory access to build histograms, where performance is also constrained by scratchpad capacity that determines bucket sizing. Modern FPGA platforms such as the V80 provide on-chip scratchpad on the order of ~100 MB, significantly larger than GPU L1 scratchpad (e.g., on NVIDIA H100), enabling more efficient implementations of such operations.
>
> > How does the GPU-FPGA heterogeneous system perform under large batch sizes?
>
> This table illustrates the batch size scaling (geomean speedup) of the GPU-FPGA systems:
>
> | Method | BS=1 | BS=2 | BS=4 | BS=8 | BS=32 |
> |---|---|---|---|---|---|
> | SA-R Thres. | 1.12x | 1.21x | 1.33x | 1.47x | 1.60x |
> | DSA | 1.02x | 1.05x | 1.09x | 1.14x | 1.15x |
> | SA-R Top-k | 1.07x | 1.15x | 1.24x | 1.30x | 1.32x |
> | LServe | 1.19x | 1.34x | 1.51x | 1.67x | 1.83x |
> | DRAGIN | 1.14x | 1.20x | 1.29x | 1.44x | 1.92x |
> | FLARE | 1.19x | 1.26x | 1.38x | 1.55x | 2.11x |
> | FS-RAG | 1.26x | 1.32x | 1.42x | 1.581x | 2.10x |
> | 2-Stage RAG | 1.16x | 1.22x | 1.28x | 1.32x | 1.37x |
> | HMT/Titans | 1.48x | 1.47x | 1.45x | 1.38x | 1.15x |
> | MemAgent | 1.85x | 1.65x | 0.93x | 0.49x | 0.13x |
>
> We observe that speedup increases with batch size for sparse attention and most RAG methods, decreases for Memory-as-Context, and becomes a slowdown for MemAgent. For detailed reasons for speedup variation across batch sizes, see Q4 for review eaft. For MemAgent, we can dynamically select the optimal configuration. For example, when BS > 2 in MemAgent, we switch to a GPU-centric mode to avoid slowdown.
>
> > What are the specific architectural or algorithmic reasons for the high variance in speedups across the different sparse attention
>
> There are two primary reasons. First, the computational patterns and data storage requirements for score computation and retrieval differ across sparse attention methods. Specifically:
>
> - DeepSeek Attention: stores all lightweight key indexing vectors to support fine-grained sparsity and uses multi-head dot-product scoring.
> - SeerAttention-R: adopts block-sparse attention and stores a single indexing vector per block.
> - LServe: uses a finer-grained block-sparse design with logical and physical pages, where each page maintains two indexing vectors (minimum and maximum).
>
> These differences lead to varying speedups, as they impose distinct storage footprints, access patterns across memory hierarchies, and scoring complexities that directly impact kernel-level performance.
>
> Second, the e2e latency improvements vary because the fraction of execution time attributed to the disaggregated stages differs across methods. Since only a subset of computations is offloaded to the FPGA, the achievable speedup depends on how much of the original latency lies within these stages, thereby affecting the overall system-level gains.
>
> > a detailed, quantitative timeline or breakdown showing the exact PCIe transfer latency
>
> A high-level comparison between the PCIe latency and the end-to-end latency is in Appendix C.1. We will include a detailed breakdown for each method in the final version. The following numbers compare the PCIe latency with the latency of the baseline for part of the computations we offloaded to FPGA:
>
> - Sparse Attention: Transfers include new key indexing vectors and retrieved indices, taking approximately 12 µs. The computations take 128–2450 µs.
> - RAG: Transfers only include retrieved indices, taking approximately 7 µs. The computations take 23–1596 ms.
> - Memory-as-Context: Transfers include memory, query, and retrieved embeddings for each segment, taking approximately 20–320 µs. The computations take 26–498 ms.
> - MemAgent: Transfers include KV cache and token IDs for each segment, taking approximately 14–218 ms. The computations take 17–534 s.
>
> These comparisons show that PCIe communication overhead remains small (~1000x difference) relative to computation time across all methods, even when data transfer size increases.

---

> > ### Author Rebuttal · Reviewer_eQks · 2026-04-02
> >
> > Thank you for your response. I have no further questions.

---

> > > ### Author Response · Authors · 2026-04-04
> > >
> > > Thank you for the time to review our work and consider our rebuttal. We appreciate your feedback.

---

### Official Review · Reviewer_tGAu · 2026-03-12

**Soundness:** 2
**Presentation:** 3
**Significance:** 2
**Originality:** 2
**Overall Recommendation:** 4
**Confidence:** 3

**Summary:**

To mitigate the high memory processing overhead in LLM inference, this paper proposes a unified framework that abstracts existing long-context processing mechanisms into a standardized four-step memory processing pipeline on heterogeneous hardware.

Through systematic profiling, the study reveals that this pipeline accounts for 22% to 97% of total inference latency and exhibits strong computational heterogeneity, mixing compute-dense operations with irregular, data-dependent, and memory-bound tasks. To address these bottlenecks, the authors design a GPU-FPGA heterogeneous system by offloading irregular, memory-bound tasks to the FPGA while keeping dense, compute-intensive operations on the GPU. Evaluations show end-to-end speedup and energy savings.

**Compliance With Llm Reviewing Policy:**

Affirmed.

**Final Justification:**

The responses from the authors addressed my concerns well. Therefore, I am changing my score to a positive one.

**Key Questions For Authors:**

- Why is there a necessity for a unified abstraction?
- Do the inherent memory limitations of FPGAs compromise the generalizability of the system relative to GPU-centric architectures? I do not see experiments under long seq_len in the evaluation.
- Clarify more on the novelty compared with other FPGA offloading-based work.
- Fix typo in L032: GPU-OSS-120B->GPT-OSS-120B

**Limitations:**

- Hardware Scalability: The strict HBM capacity limits of the chosen FPGA, and how the system performs under long seq_len.

- PCIe Bottlenecks: While PCIe is sufficient for index transfers, it remains a severe bottleneck for larger tensor transfers, making the system highly dependent on the host CPU's PCIe topology for this heterogeneous system.

- Deployment Friction: The engineering overhead of developing custom FPGA kernels (HLS) compared to writing standard GPU kernels, and the potential lack of portability across different hardware vendors.

**Strengths And Weaknesses:**

## Strengths

- The paper presents an empirical evaluation on physical hardware (AMD MI210 GPU and Alveo U55C FPGA) rather than simulation only. It proves e2e speedups and energy savings across diverse state-of-the-art workloads like DeepSeek Attention and LServe.
- The narrative is driven by an exceptionally clear and uniform abstraction: unifying disparate long-context methods (RAG, sparse attention, compressed memory) into a single, easily digestible four-step "memory processing pipeline".
- The work demonstrates energy and latency reductions according to the evaluation.

## Weaknesses

- As stated by the authors in Sec 6.1, the hardware comparison pits an older 16nm FPGA against a newer 6nm GPU. The differences in both performance and cost slightly skew the architectural baseline fairness.
- Given the significant disparities between the four mechanisms discussed in Section 3.1 (skip a few steps), the necessity of a unified abstraction is not entirely clear. Specifically, since several methods require skipping intermediate steps to fit the model, would it be more efficient to maintain specialized systems? The authors should justify whether this unification introduces unnecessary complexity or performance overhead compared to standalone implementations.
- The core strategy of offloading specific LLM components to FPGAs has been explored (FlightLLM, LUT-LLM).

---

> ### Author Rebuttal · Authors · 2026-03-29
>
> Thank you for your detailed review and feedback. We respond to the weakness, key questions, and the limitations as follows:
>
> > The hardware comparison pits an older 16nm FPGA against a newer 6nm GPU.
>
> It is worth noting that using an FPGA with an older node than the GPU makes the speedup and energy improvement more challenging, and it further shows how significant our work is. More recent FPGAs, such as AMD V80, provide significantly higher compute resources and HBM bandwidth. Based on kernel synthesis results, we estimate an additional 1.6x geomean end-to-end speedup when migrating to V80.
>
>
> > Why is there a necessity for a unified abstraction?
>
> From the weakness section, we consider that your question refers to the methods summarized in Table 1, particularly regarding the necessity of a unified abstraction:
>
> - Only 2 out of the 9 methods omit certain stages, and each skips a different step. This variability suggests that retaining all stages in the memory processing pipeline is necessary to provide a complete and general abstraction that covers diverse designs.
>
> - The memory processing pipeline serves primarily as an analytical framework to characterize heterogeneity and guide disaggregation. When a stage is not required, it introduces no overhead, as data can bypass the stage without additional computation or control complexity.
>
> - A unified abstraction not only enables systematic analysis of disaggregation strategies but also supports reuse of kernel designs and deployment strategies across methods that exhibit similar heterogeneity patterns.
>
> > The core strategy of offloading specific LLM components to FPGAs has been explored (FlightLLM, LUT-LLM).
>
> Both FlightLLM and LUT-LLM implement LLM inference entirely on FPGA as standalone systems and do not consider GPU–FPGA offloading. In contrast, prior work on GPU–FPGA disaggregation either lacks real-system deployment, as in GLITCHES@HPEC’24, or focuses on optimizing a single method, such as CXL-SpecKV@FPGA’26 and Chameleon@VLDB’24. Our approach instead provides a systematic methodology to determine which components of LLM inference are well-suited for FPGA execution and how they should be integrated with GPU computations, with the consideration of cross-device communication overheads. This enables consistent performance gains across a broad range of methods rather than being tailored to a specific optimization. Furthermore, the design is not restricted to GPU-FPGA systems; the memory processing pipeline abstraction and its associated deployment strategy generalize naturally to other heterogeneous platforms.
>
>
> > I do not see experiments under long seq_len in the evaluation.
>
> The long seq_len setting can be found in Figure 8-12 of Section 6, scaling the sequence to 1M tokens for sparse attention and 20M documents for RAG.
>
> > While PCIe is sufficient for index transfers, it remains a severe bottleneck for larger tensor transfers.
>
> We will include a detailed breakdown for each method in the final version. At a high level, “large tensor transfer” does not become the bottleneck because methods that require larger data movement also exhibit substantially longer end-to-end latency, making PCIe overhead relatively small in comparison (See Appendix C.1). The following numbers compare the PCIe latency with the latency of the baseline for part of the computations we offloaded to the FPGA:
>
> - Sparse Attention: Transfers include new key indexing vectors and retrieved indices, taking approximately 12 µs. The computations take 128–2450 µs.
> - RAG: Transfers only include retrieved indices, taking approximately 7 µs. The computations take 23–1596 ms.
> - Memory-as-Context: Transfers include memory, query, and retrieved embeddings for each segment, taking approximately 20–320 µs. The computations take 26–498 ms.
> - MemAgent: Transfers include KV cache and token IDs for each segment, taking approximately 14–218 ms. The computations take 17–534 s.
>
> These comparisons show that PCIe communication overhead remains small (~1000x difference) relative to computation time across all methods, even when data transfer size increases.
>
>
> > The engineering overhead of developing custom FPGA kernels
>
> Thank you for raising this limitation, and we will mention it as a limitation and one of our future work in the final version. We would also like to mention that many existing works either support a high-level interface for fast FPGA design (Allo@PLDI’25, StreamHLS@FPGA’25), speed up RTL simulation (LightningSim@FCCM’23), or accelerate the kernel bitstream generation backend (RapidStream@FPGA’23). These works shrink the gap in development efforts between FPGA kernels and GPU kernels.

---

> > ### Author Rebuttal · Reviewer_tGAu · 2026-04-02
> >
> > Thank you for your response. I have no further questions.
> >
> > I suggest including the clarification of the unified abstraction and highlighting the PCIe overhead in the corresponding main sections, if possible.

---

> > > ### Author Response · Authors · 2026-04-04
> > >
> > > Thank you for taking the time to review our work and consider our rebuttal. We will incorporate your suggestions in the final version, including clarifying the unified abstraction, moving the PCIe overhead discussion to the main paper, and addressing the typos. We sincerely appreciate your feedback.

---

### Official Review · Reviewer_Q6ho · 2026-03-12

**Soundness:** 3
**Presentation:** 3
**Significance:** 3
**Originality:** 3
**Overall Recommendation:** 5
**Confidence:** 3

**Summary:**

This paper analyzed the bottle neck of modern llm inference, and explain how the memory access is the common bottle neck accross architecture. The authors then proposes a GPU-FPGA heterogeneous system that accelerates LLM inference by unifying diverse memory optimization methods into a four-step pipeline and strategically dispatching memory-bound, irregular operations to FPGAs, and rest for gpu.

**Compliance With Llm Reviewing Policy:**

Affirmed.

**Key Questions For Authors:**

no

**Strengths And Weaknesses:**

strength

1. solid analysis: the four stage convers a wide range of popular/novel architectures, showing the generalization of such analysis framework.

2. altough offloading sparse memory ops to fgpas is not very novel, but the fine-grained optimization, especially in relevancy/retrieval stage is a novel contribution.

3. extensive experiments and promissing results. a hybrid serving hardwares offer flexibility to hardware types and cost and can also provide speed up overall.

4. nice discussion at the end.

---

> ### Author Rebuttal · Authors · 2026-03-31
>
> Thank you for the positive feedback and for recognizing the contributions of our work. We have no further clarifications.

---

### Official Review · Reviewer_eaft · 2026-03-13

**Soundness:** 3
**Presentation:** 3
**Significance:** 2
**Originality:** 3
**Overall Recommendation:** 4
**Confidence:** 3

**Summary:**

This paper proposes that diverse LLM inference optimizations (sparse attention, RAG, compressed memory, TTT) share a common four-step memory processing pipeline. The computational characteristics of these steps are heterogeneous. The authors validate that a
GPU-FPGA system can exploit this heterogeneity to accelerate inference on an AMD MI210 + Alveo U55C platform, achieving 1.04-2.2X speedup and 1.11-4.7X energy reduction over a GPU-only baseline.

**Compliance With Llm Reviewing Policy:**

Affirmed.

**Final Justification:**

Thank you for the detailed responses. The A100 estimation methodology is well-justified given the absence of a co-located A100+U55C platform, and the <3µs error validation on MI210 is reassuring. The batch scaling table directly addresses my concern about concurrency, and the PCIe breakdown confirms the communication overhead is negligible relative to computation time. The V80 projection and cost analysis strengthen the practical case for modest speedups on current hardware. I will consider adjusting my score accordingly.

**Key Questions For Authors:**

See weaknesses.

**Limitations:**

Yes.

**Strengths And Weaknesses:**

Strengths:
* This paper introduces a unified pipeline abstraction. I believe this is the core contribution.
* The evaluation is very comprehensive. It covers DeepSeek Attention, SeerAttention-R, LServe, three RAG variants, MemAgent, and Memory-as-Context.

Weaknesses:
* The A100 evaluation is estimated, not measured.
* For sparse attention, many end-to-end speedups are in the 1.04-1.2X range, meaningful but not compelling given the engineering complexity of deploying and maintaining FPGA kernels. The paper should address the question of whether a simpler CPU offload approach would achieve similar gains with far less development overhead.
* The evaluation primarily scales sequence length but largely ignores massive batched concurrency.

---

> ### Author Rebuttal · Authors · 2026-03-29
>
> Thank you for your detailed review and feedback. Our responses are the following:
>
> > The A100 evaluation is estimated, not measured.
>
> First, no publicly accessible platform integrates A100 and U55C within the same node. Second, the A100+U55C results are not derived from simulations, but **based on real-system measurements**. Concretely, we first profile execution on an A100-only system to obtain a latency breakdown, then reconstruct e2e latency by replacing the FPGA-offloaded components with measured U55C kernel latency and PCIe communication overhead. Because MI210, A100, and U55C all operate over PCIe Gen4, the inter-device latency characteristics are consistent, ensuring that the estimation is close to real-system deployment. To validate this methodology, we apply the same method to the MI210+U55C system and compare against our e2e measurements, observing an error within 3 us.
>
> > many end-to-end speedups are in the 1.04-1.2X range, meaningful but not compelling.
>
> We add the following clarifications to strengthen the practical impact of the reported speedups:
>
> - The U55C FPGA is built on a 16nm process, which is two generations older than the 7nm technology used in A100. More recent FPGA platforms, such as AMD V80 provide significantly higher compute resources and HBM bandwidth. Based on kernel synthesis results, we estimate an additional 1.6x geomean e2e speedup when migrating to V80. The kernel design paradigms remain unchanged, making this transition straightforward.
>
> - In production, serving cost is equally critical. The FPGA kernels exhibit substantially lower power consumption (Appendix G), resulting in up to 1.6x lower overall energy cost for sparse attention. Furthermore, FPGA instances can also be more cost-effective than CPU-based offloading. For example, replacing an AWS EC2 m7a.24xlarge instance, which has comparable CPU and memory resources to our baseline system, with an AWS EC2 f2.6xlarge instance equipped with FPGA resources similar to U55C can **reduce hourly cost by $3.6 (which is 2.8x reduction)** while still achieving performance improvements.
>
> > whether a simpler CPU offload approach would achieve similar gains with far less development overhead.
>
> We clarify that RAG baselines already offload portions of computation to the CPU (e.g., BM25 scoring), as dictated by underlying baseline frameworks and implementations. For sparse attention, memory-as-context, and MemAgent, CPUs offer lower peak throughput and depend on relatively low-bandwidth DRAM, which constrains performance for memory-bound, irregular workloads (See Appendix F).
>
>
> > The evaluation primarily scales sequence length but largely ignores massive batched concurrency
>
> The following table illustrates the batch size scaling (geomean speedup) of the GPU-FPGA systems over all the methods:
>
> | Method | BS=1 | BS=2 | BS=4 | BS=8 | BS=32 |
> |---|---|---|---|---|---|
> | SA-R Thres. | 1.12x | 1.21x | 1.33x | 1.47x | 1.60x |
> | DSA | 1.02x | 1.05x | 1.09x | 1.14x | 1.15x |
> | SA-R Top-k | 1.07x | 1.15x | 1.24x | 1.30x | 1.32x |
> | LServe | 1.19x | 1.34x | 1.51x | 1.67x | 1.83x |
> | DRAGIN | 1.14x | 1.20x | 1.29x | 1.44x | 1.92x |
> | FLARE | 1.19x | 1.26x | 1.38x | 1.55x | 2.11x |
> | FS-RAG | 1.26x | 1.32x | 1.42x | 1.581x | 2.10x |
> | 2-Stage RAG | 1.16x | 1.22x | 1.28x | 1.32x | 1.37x |
> | HMT/Titans | 1.48x | 1.47x | 1.45x | 1.38x | 1.15x |
> | MemAgent | 1.85x | 1.65x | 0.93x | 0.49x | 0.13x |
>
> We observe that speedup increases with batch size for sparse attention and most RAG methods, decreases for Memory-as-Context, and becomes a slowdown for MemAgent:
>
> - **Sparse attention.** KV cache or indexing embeddings are not shared across samples (Atom@MLSys’24), so batching does not improve their data reuse or GPU parallelism for score computation while GPU+FPGA can still exploit the high memory utilization advantages. In contrast, dense operations (e.g., projections, FFN) benefit from weight reuse. This increases the latency fraction of sparse attention, amplifying the benefit of offloading and yielding higher speedup.
>
> - **RAG.** DRAGIN, FLARE, and FS-RAG use BM25 retrieval that data access is input-dependent. The reason for the increase of speedup is similar to sparse attention. Two-stage RAG partially offsets this trend due to improved reranker efficiency under batching.
>
> - **HMT/Titans.** The linear projections in the cross attention benefit from batching via weight reuse and higher GPU utilization. However, memory embeddings remain sample-independent, so FPGA acceleration still helps in long-sequence regimes.
>
> - **MemAgent.** The memory processing is standard LLM inference, where batching significantly improves GPU efficiency in decode. Due to lower FPGA throughput for dense operations, this leads to slowdown as batch size increases.
>
> For MAC and MemAgent, we can dynamically select the optimal configuration. For example, when BS > 2 in MemAgent, we switch to a GPU-centric mode to avoid slowdown.

---

> > ### Author Rebuttal · Reviewer_eaft · 2026-04-02
> >
> > Thank you for the detailed responses. The A100 estimation methodology is well-justified given the absence of a co-located A100+U55C platform, and the <3µs error validation on MI210 is reassuring. The batch scaling table directly addresses my concern about concurrency, and the PCIe breakdown confirms the communication overhead is negligible relative to computation time. The V80 projection and cost analysis strengthen the practical case for modest speedups on current hardware. I will adjust my score accordingly.

---

> > > ### Author Response · Authors · 2026-04-04
> > >
> > > Thank you for the time to review our work and consider our rebuttal. We appreciate your feedback.

---

### Decision · Program_Chairs · 2026-04-30

**Decision:**

Accept (regular)

**Comment:**

The reviews are broadly positive. Reviewers agree that the paper’s main strength is its clear four-stage memory-processing abstraction, which unifies several LLM inference optimization settings and motivates a sensible GPU-FPGA partitioning. They also find the empirical study broad and solid, with meaningful latency and especially energy improvements across diverse workloads.

The main concerns were about generalization to newer hardware, reliance on estimated rather than fully measured cross-platform results, limited batching/high-throughput analysis, and clearer positioning against prior FPGA offloading work. The authors’ rebuttal addressed these points well by clarifying the abstraction’s role, providing batching and PCIe-overhead analysis, and strengthening the discussion of practicality and novelty. Multiple reviewers explicitly stated that their concerns were fully resolved.

Overall, the paper is viewed as technically solid, well presented, and likely to be useful to the systems/ML hardware community, though its impact is moderated by some remaining questions about deployment on newer platforms and real-world serving settings.